# Chiral plasmonic superlattices from template-assisted assembly of achiral nanoparticles

Xiaoyu Qi, Luis Alberto Pérez ✉, Jose Mendoza-Carreño, Miquel Garriga, Maria Isabel Alonso & Agustín Mihi ✉

The creation of chiral plasmonic architectures combining templates with achiral plasmonic particles leads to strong chiroptical responses that can be finely tuned via the characteristics of the colloidal building blocks. Here we show how elastomeric molds, pre-patterned with a hexagonal array of triskelia motifs, can guide the assembly of ordinary noble metal colloids into chiral plasmonic architectures with strong dichroism values. Under normal incidence, the chiral arrays made with gold and silver colloids showed $g$-factors of 0.18 and 0.4, respectively. In all cases, increasing the size of the colloid allows tuning the optical properties of the structure in the VIS-NIR range. When a superstrate layer is deposited onto the structures, the extrinsic chirality response of the 2D superlattice is revealed and strongly amplified by the chiral motifs under oblique inspection, leading to $g$-factors of $\pm 1.2$ at $\pm 14°$. Finally, these chiral plasmonic resonances sustained by the triskelion array are used to produce circularly polarized photoluminescence from achiral organic dyes placed on top with up to 20% of dissymmetry.

The captivating phenomenon of chirality manifests in various aspects of nature, including fundamental life components like proteins, lipids and amino acids[1–3]. In chemistry, an atom bound to four different chemical species acts as a chiral center, thus enabling the existence of left- and right-handed versions of the same molecule, the so-called enantiomers[4]. Although enantiomers have the same chemical composition, they may have dramatically different effects on living organisms or the environment, triggering the great interest in their differentiation[5,6]. Chiral molecules present slightly different dielectric response for left- or right-handed circularly polarized light, which is exploited to distinguish enantiomers via optical spectroscopy[7–9]. Circular dichroism (CD) quantifies the difference in absorption between left- and right-handed light when passing through chiral media. These CD values can be expressed normalized to the total absorption, resulting in a well-known parameter: the $g$-factor or Kuhn's dissymmetry factor[10].

$$g-\text{factor} = \frac{\text{CD}}{E} = 2\frac{(E_{\text{LCP}} - E_{\text{RCP}})}{E_{\text{LCP}} + E_{\text{RCP}}} \qquad (1)$$

Where $E_{\text{LCP}}$, $E_{\text{RCP}}$ and $E$ are the sample extinction when illuminated with left circular polarized (LCP), right circular polarized (RCP) and non-polarized light respectively.

Most chiro-optical responses found in nature are weak due to the fundamental mismatch in size between the biomolecules and the wavelength of incident light[8,11]. However, advances in nanofabrication have led to the development of artificial photonic structures that can exhibit a strong chiroptical response engineered across the visible and near infrared spectrum[12,13]. This is the case of chiral plasmonic structures containing nanostructured metals with clever geometrical designs sustaining polarization dependent plasmonic resonances[14–16].

Chiral plasmonic architectures imply an important development in light-matter interaction, sustaining properties such as negative refractive index or non-reciprocal light propagation[17–19]. However, the intricate geometries required to support chiral optical resonances pose overarching fabrication challenges which might hinder further commercial application. The first examples of chiral nanostructures operated mainly in the NIR and were fabricated via conventional electron

Institute of Materials Science of Barcelona ICMAB-CSIC; Campus UAB, Bellaterra, Spain. ✉e-mail: lperez@icmab.es; amihi@icmab.es

beam lithography[20–25] followed by several attempts to develop alternative, simpler fabrication strategies[23,26–28]. These chiral architectures have recently shown great potential for sensing applications[29–31].

Still, there is a great interest in producing nanostructures that operate in the visible, but also, whose production can be easily scaled up[32]. This is where the chemical growth of metal colloids has played a pivotal role, bridging the gap between laboratory and application[33,34]. Recently, chiral metal colloids have been developed: González-Rubio et al. demonstrated that certain surfactants could promote a chiral morphology during growth of gold nanorods, reaching a g-factor of 0.2[35]. Lee et al. used amino acids and peptides on gold cubes leading to gammadion-like shaped particles reaching similar g-factor values[36]. Recent studies show that the values of dichroism sustained by chiral metal colloids can be even higher if they are cleverly arranged on a substrate[29,37]. Alternatively, the more easily produced spherical or rod-like gold colloids have been endowed with chiral properties thanks to the use of liquid crystal nanotubes[38], molecular templates like DNA assembly or chiral nanotubes, reaching g-factor values as high as 0.12[39].

In this work, we design, fabricate and characterize chiral arrays composed of achiral noble metal colloids, i.e., nanospheres and nanorods. The fabrication process relies on template-assisted self-assembly[40,41]. In this soft lithography process, pre-patterned elastomeric molds induce the ordering of concentrated colloidal dispersions, resulting in ordered plasmonic films. This technique has proven effective for obtaining colloidal plasmonic arrays with notable optical features[42–45]. Here we expand the use of this scalable nanofabrication technique to produce chiral plasmonic architectures. The elastomeric molds used herein feature hexagonal arrays of left- or right-handed triskelia designed to maximize the CD of the geometry. This method applied to different noble metal colloids allowed the production of plasmonic chiral films with g-factors of 0.18 for Au nanospheres and 0.4 for Ag colloids under normal incidence. This chiral lattice design allows the contribution from the chiral lattice under oblique observation across a wide angular range, spanning from 7° to 45° (within the wavelength range of 750 to 950 nm) reaching a maximum g-factor of 1.2 at 14° when using a refractive index matching oil superstrate (and g-factors of 1.55 under an epoxy coating). The combination of chiral geometry and scalable template-assisted assembly is employed herein to endow chirality to silver and gold nanospheres and gold nanorods. Furthermore, we demonstrate how the chiroptical response of these chiral plasmonic arrays can be used to obtain circularly polarized photoluminescence (CPL) from a non-chiral NIR dye placed on top with values that can be actively tuned in terms of sign, magnitude and wavelength by adjusting the collecting angle of the light.

## Results

### Design and fabrication of triskelion arrays

The chiral patterns were designed following two main considerations; the first was that the geometry had to be easily replicated by soft-nanoimprinting lithography and the second was that it had to maximize the CD from the structure. The triskelion motif was selected due to its inherent threefold symmetry[46], which aligns well with the desired chiral properties and is an open structure that can be replicated in elastomeric molds. Optimization of the geometry was achieved through electrodynamic modeling utilizing the finite-difference time-domain (FDTD) method and comparing the values of CD obtained in each case. The hexagonal lattice was chosen to maintain the threefold symmetry of the pattern. FDTD simulations confirmed that the hexagonal lattice pattern yields higher CD and lower linear dichroism (LD) compared to a triskelia square lattice. (Supplementary Fig. 1) In the final design, each triskelion is formed by three curved radii, with a distance of 400 nm between vertices, the width of each arm is 75 nm, and the lattice parameter (LP) of the hexagonal lattice is 500 nm. The final chiral array (Fig. 1a) was engraved on a silicon wafer by means of conventional lithography, serving as the original master to produce the left- or right- patterned printing polydimethylsiloxane (PDMS) stamps (see "Methods") used for the template assembly of the metal colloids.

The template-assisted fabrication of triskelion arrays with gold colloids is outlined in Fig. 1b. Briefly, PDMS printing stamps pre-patterned with a 2D hexagonal array of left- or right-handed triskelion cavities, covering a 16 mm² area, were utilized to guide the assembly of the colloids[44,45]. Gold and silver colloids were functionalized with Poly(ethylene glycol) methyl ether thiol (PEG-SH, $M_w$ = 2000 g mol⁻¹) dispersed in a water: ethanol (6:4) mixture, and stabilized with hexadecyltrimethylammonium chloride (CTAC), see "Methods". A 1.5 μL volume of the colloidal dispersion was placed onto cleaned glass (20 × 20 mm²) and covered with the PDMS stamps. After complete solvent evaporation (4–6 h), the stamp was removed leaving a plasmonic array on the glass.

We compared the values of CD predicted by FDTD simulations in triskelion arrays made of solid Au versus those made with discrete Au nanospheres. As illustrated in Fig. 1c, d, the calculated g-factor spectra for triskelion architectures composed of spherical Au nanoparticles exhibit significantly stronger chiroptical activity compared to an equivalent structure fabricated from bulk gold. Notably, the simulated spectrum of g-factor versus wavelength greatly resembles the experimental results obtained for 27 nm Au nanospheres (Fig. 1e). The transmittance spectra alongside the corresponding CD and LD spectra for the calculated structures is presented in Supplementary Fig. 2. These results demonstrate that the nanoparticle-based architecture achieves ellipticity values close to 8°, surpassing the 2° observed for the bulk counterpart. Furthermore, a 35-fold contrast is observed between the chiroptical response and LD at the wavelength corresponding to the maximum response. In sum, the colloid-made chiral structures not only show a straightforward fabrication process, but also show a higher degree of dichroism compared to the bulk counterparts. In addition, we compared the chiroptical properties of different chiral architectures fabricated using the same colloidal nanoparticles and found that the triskelion arrays provided the best results (see Supplementary Figs. 15, 16).

Next, we investigate the nature of the chiral optical resonances sustained by the triskelia. An isolated gold triskelion on a glass substrate is expected to sustain a non-uniform spatial distribution of the electric field when illuminated by circularly polarized light of $\lambda$ = 690 nm as predicted by FDTD simulations (Fig. 1f). The electric field is concentrated primarily on the inner side of the arms and tips of the triskelion. The template-assisted assembly of metal colloids allows for the preparation of films on top of transmission electron microscopy (TEM) grids (Fig. 1h) where electron energy loss spectroscopy (EELS) can be performed (see Supplementary Fig. 6). The EELS maps measured in chiral arrays fabricated with 27 nm Au NPs showed similar field distributions to those predicted by the FDTD model (Fig. 1g). The EELS map within the range of 1.8–1.9 eV primarily localizes losses on the inner parts of the arms, indicative of the plasmonic collective behavior of nanoparticles within the triskelion, exhibiting plasmonic modes with distributions resembling the chiral field distributions found in the FDTD simulations[47].

### Structural characterization of chiral plasmonic arrays

Template-assisted self-assembly leads to hexagonal arrays of triskelia formed by gold or silver nanospheres over 16 mm² areas as illustrated in the low-magnification scanning electron microscope (SEM) image in Fig. 2a. The inset photograph shows the colorful diffraction observed with the naked eye from the chiral lattice. The versatile fabrication process is compatible with many types of materials which was exploited herein to study how the size, material and shape of the colloids influence the optical response of the nanostructure. In particular, colloidal nanospheres of gold with diameters of 27, 45 and 54 nm and of silver with diameters of 20, 27, 32 and 36 nm along with gold

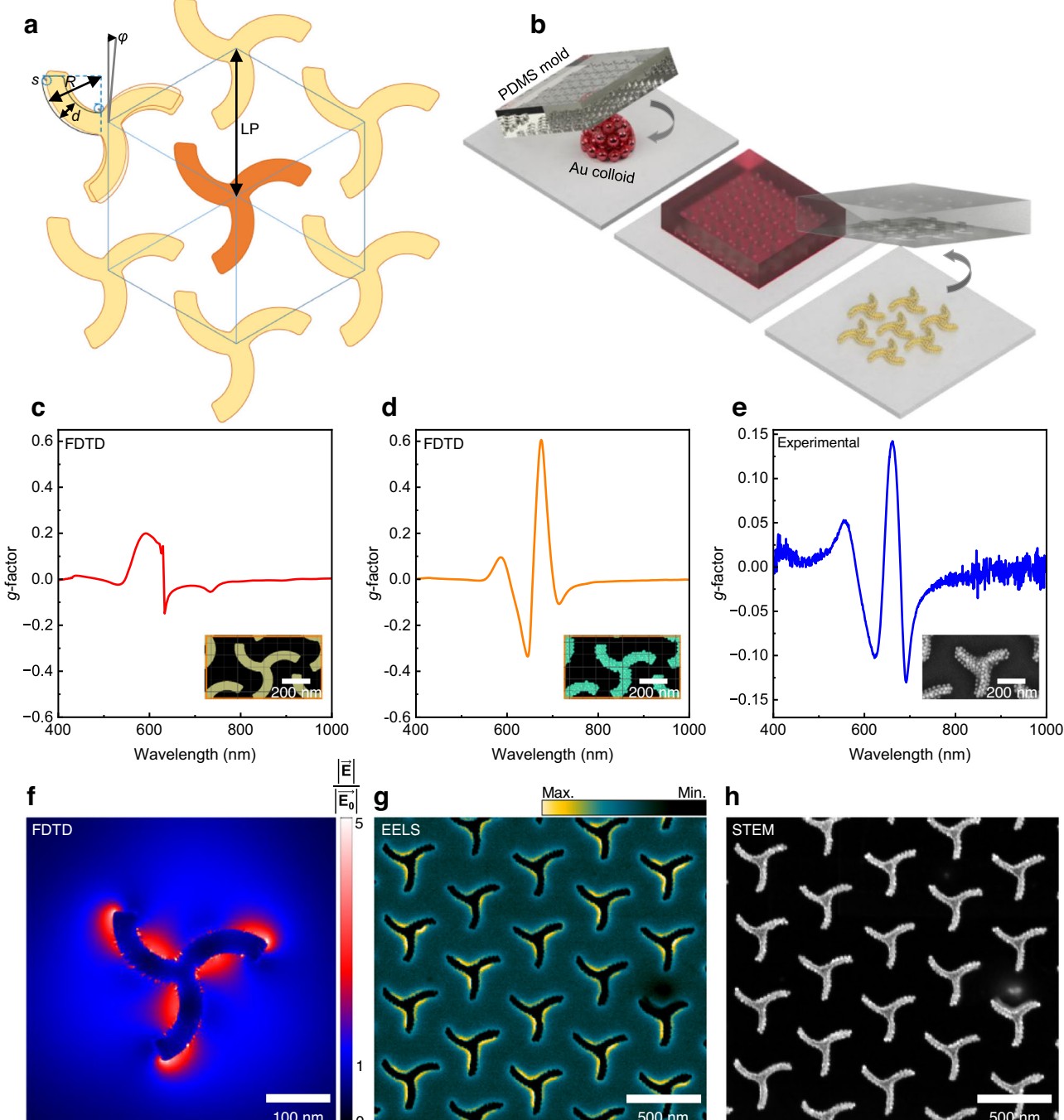

**Fig. 1 | Chiral plasmonic arrays design, gold colloids assembly, FDTD calculations, experimental *g*-factor and EELS characterization. a** Schematic representation of the designed single triskelion pattern and arranged in a hexagonal lattice. R = 200 nm, d = 75 nm, s = 15 nm, φ = 5°, LP = 500 nm. **b** Schematic representation of the fabrication process. FDTD calculated *g*-factor spectra for (**c**) 100 nm height bulk Au triskelia array, (**d**) modeled triskelia array composed of Au nanoparticles, and (**e**) experimental *g*-factor spectrum for 27 nm Au nanoparticles triskelia assembly. The insets in **c, d** depict the modeled structures. Whereas in **e**, an SEM image of the sample is shown. **f** Cross-section of the FDTD simulated electric field distribution for a single bulk Au triskelion. **g** EELS map in the energy range of 1.8–1.9 eV. **h** STEM image of the triskelion array on a TEM grid. Source data are provided as a Source Data file.

nanorods 60 nm × 15 nm in size were used to form different sets of chiral plasmonic arrays. Figure 2 shows representative SEM images from L-handed arrays of triskelia made with 27 nm silver colloids, 45 nm gold nanospheres and 60 nm × 15 nm gold nanorods at the optimized concentrations of 50 mM silver ([Ag⁰]) or gold ([Au⁰]). The SEM images in Fig. 2 illustrate the quality of the 2D arrays of triskelia achieved with the different colloidal inks. The colloid concentration was optimized to produce clean areas of well-defined triskelia

(Supplementary Figs. 10–12). Let us briefly address the effects of imperfections such as missing arms, random particles, and size variations, which generally lead to a reduction in obtained *g*-factors. For example, particles located outside the triskelion structure contribute to the overall extinction but do not participate in the chiral response. As a result, the denominator in the *g*-factor equation increases while the numerator remains unchanged, causing a decrease in the calculated *g*-factor. To mitigate these effects, substantial efforts have been

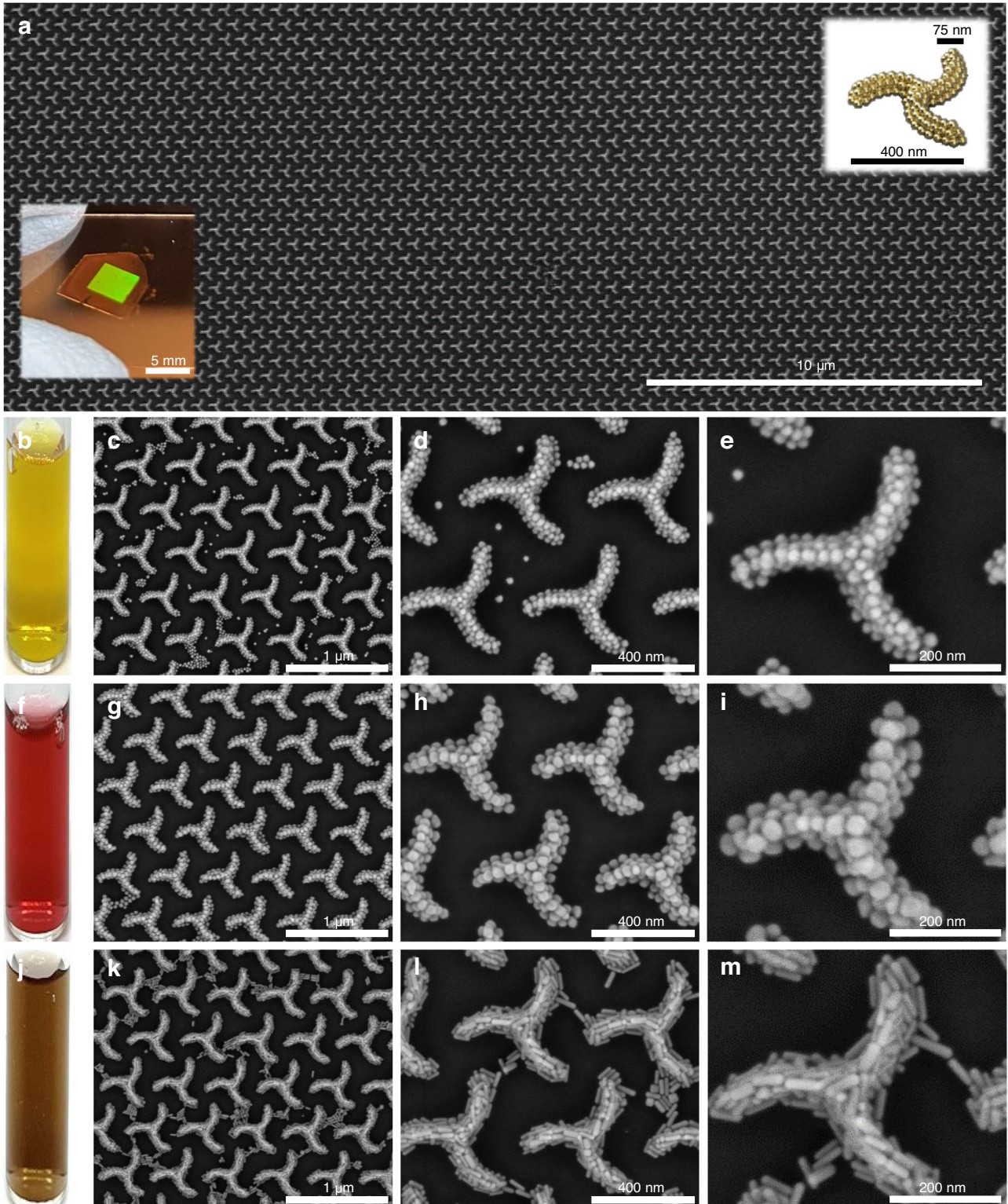

**Fig. 2 | SEM images of chiral plasmonic arrays. a** Low-magnification SEM image of the resulting 2D triskelia lattice. Insets: Photograph of a representative sample showing diffraction from the 16 mm² patterned area and a schematically representation of the triskelion unit constituted by metal nanospheres. **b**, **f** and **j** corresponds to photographs of each colloid in dispersion used to produce the different nanostructures. SEM images of the chiral arrays produced with (**c–e**) 27 nm Ag spheres, (**g–i**) 45 nm Au spheres and (**k–m**) Au nanorods (60 nm × 15 nm).

invested in refining our experimental protocols to produce arrays with low defect density (see "Methods").

For additional SEM images from other nanoparticle sizes, concentrations for both handedness, as well as photographs of the samples, see Supplementary Figs. 5–14.

## Optical characterization of the chiral plasmonic arrays

The optical characterization of the triskelia arrays consisted in measuring zeroth-order transmittance and CD represented by the dissymmetry factor (g-factor) (Fig. 3). For detailed descriptions of the experimental setups see Supplementary Fig. 17. All plasmonic

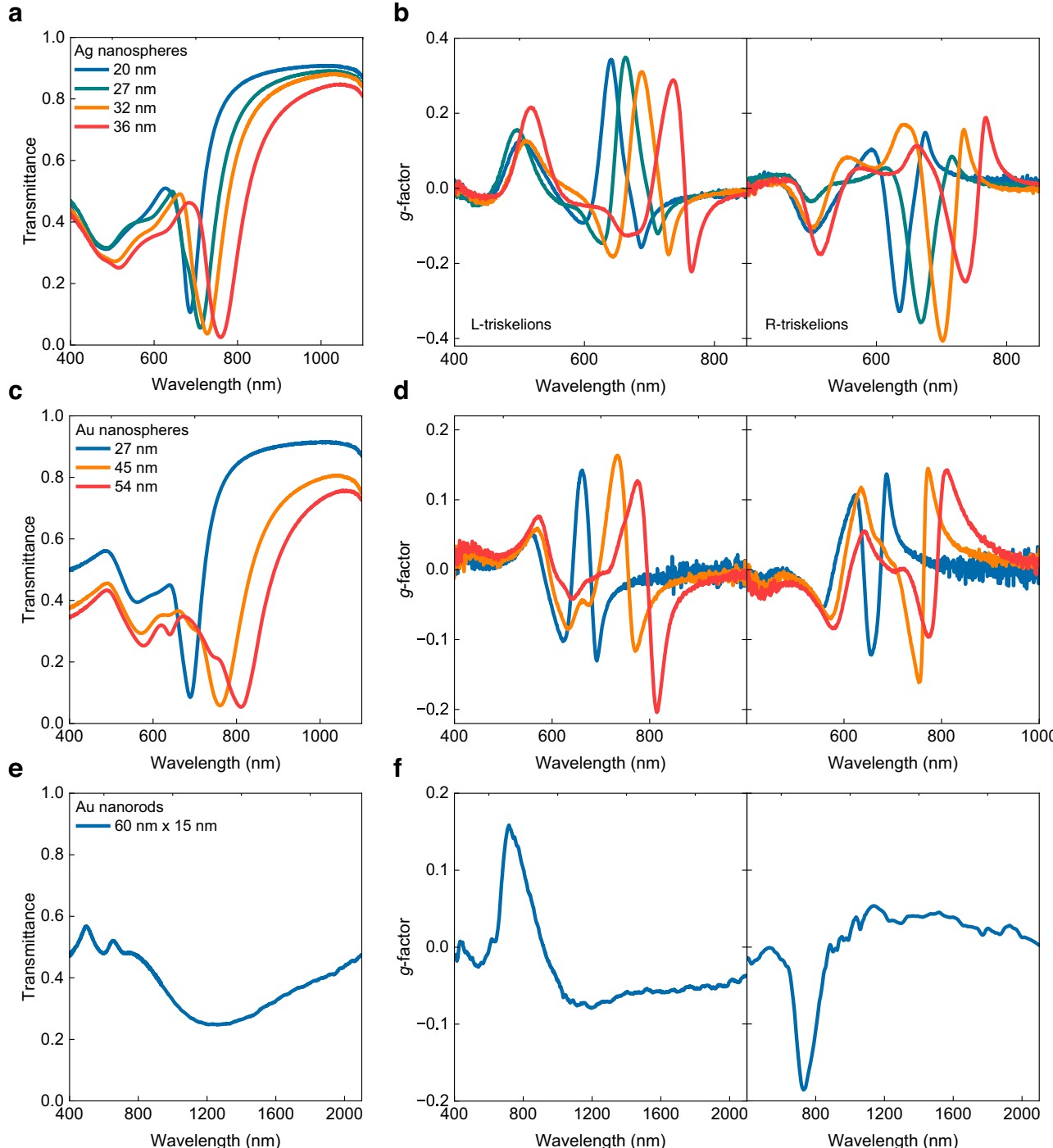

**Fig. 3 | Optical response of metal colloid triskelion patterns.** Non-polarized light transmittance and *g*-factor for left- and right-handed triskelion arrays of (**a** and **b**) Ag spheres (20 (blue lines), 27 (green lines), 32 (orange lines) and 36 nm (red lines)), (**c** and **d**) Au spheres (27 (blue lines), 45 (orange lines) and 54 nm (red lines)) and (**e** and **f**) Au rods (60 nm × 15 nm in size) (blue lines). Source data are provided as a Source Data file.

superlattices contain the same triskelia array dimensions, dictated by the PDMS molds, and differ only in the type and size of the metal colloids used in each case.

The transmittance spectra of triskelia arrays made of metallic nanospheres (Ag or Au) show a characteristic transmittance dip associated with the surface lattice resonance (SLR) between 660-800 nm. The SLR is originated from the collective excitation of the localized modes of the triskelion via the diffraction of the 2D hexagonal lattice. Towards shorter wavelengths (500-700 nm region), we find the localized surface plasmon resonance (LSPR) associated

with the triskelion shape (see Supplementary Fig. 3). Figure 3 shows the optical response of the triskelia arrays under normal incidence for all colloids whereas the full angular dispersion of the structure is presented in Fig. 4 for the silver nanoparticle case. Additional spectra from the samples can be found in Supplementary Figs. 20–22. Increasing the size of the metal colloids composing the triskelion leads to a redshift of the SLR resonance for both Ag and Au (Fig. 3a, c), which is a straightforward way to tune the optical response (both *T* and *g*-factor) of the assemblies for a given PDMS pre-patterned geometry.

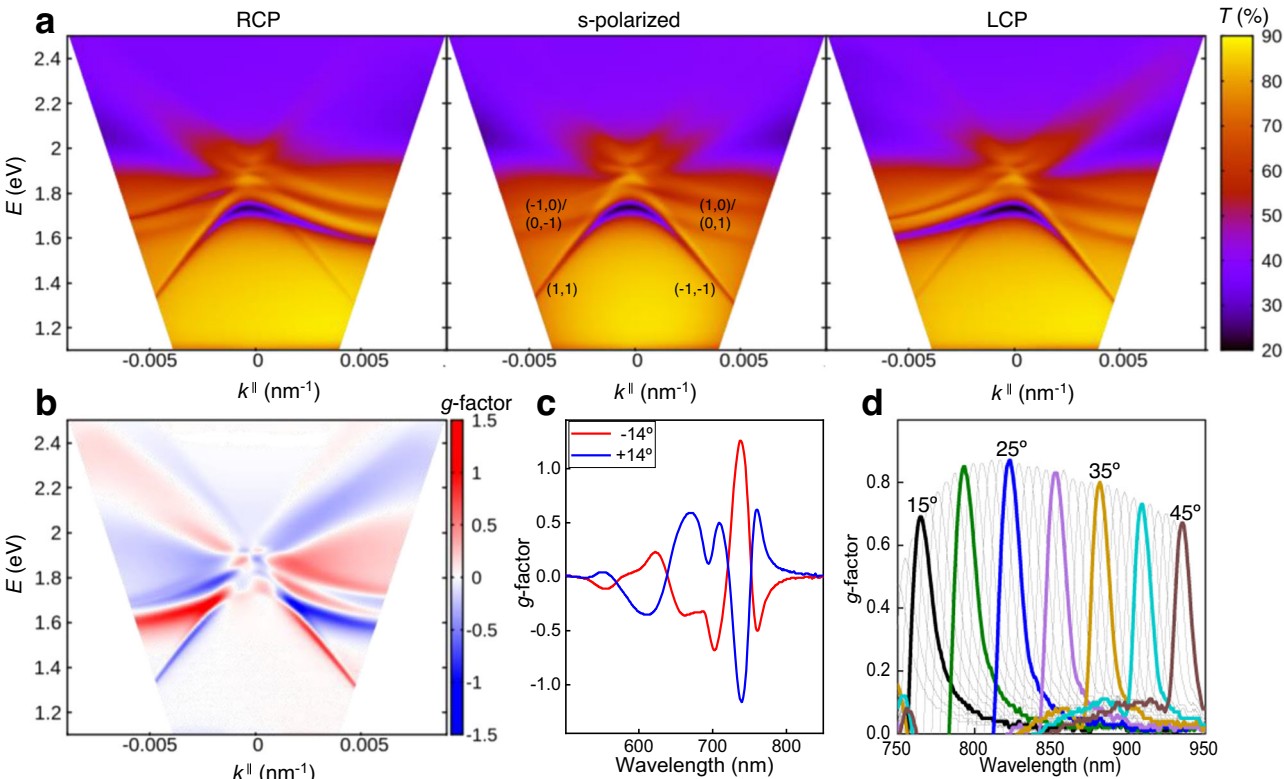

**Fig. 4 | Experimental angle-resolved transmittance for triskelion arrays made with 27 nm Ag colloids. a** Transmittance angular dispersion (from -45° to +45°) collected under RCP, linear s-polarized light and LCP in energy $E$ (eV) versus parallel wavevector $k$ (nm$^{-1}$). **b** Angle-resolved $g$-factor map. **c** Maximum $g$-factor values measured for left- handed triskelion arrays at ±14° angle of incidence. (−14° red line, +14° blue line) (**d**) $g$-factor spectra measured over an angular range of 12° to 45° in the region of 750 to 950 nm. Source data are provided as a Source Data file.

All samples showed the most intense $g$-factor values around the SLR, reaching $g$-factors of -0.18 for 54 nm Au spheres (Fig. 3d) and -0.4 for 32 nm Ag colloids (Fig. 3b). These $g$-factors are similar to those reported for gold chiral colloids ($g$-factor = 0.2) and we have been able to endow chirality to Ag colloids with a $g$-factor up to 0.4, for which no synthetic approach has been developed so far. The invariance of the optical response under reverse illumination and multiple azimuthal angles reveals the intrinsic chiral behavior of the triskelia nanostructures under normal incidence (Supplementary Figs. 24, 25)[48]. The triskelion arrays made from Au nanorods, despite showing high structural quality (Fig. 2k–m), showed broad SLR towards the NIR region (Fig. 3e). The $g$-factor calculated for the corresponding left- and right-handed triskelion arrays showed two different regions with opposite chirality, one sharp $g$-factor peak ($g$-factor of 0.16 (L) and -0.18 (R)) around 705 nm and a rather broad band ($g$-factor of -0.07 to 0.05) in the 850–1600 nm region (Fig. 3f).

In summary, increasing the size of the colloids induces a red shift of the SLR position and the CD properties of the triskelion arrays from 650 to 750 nm for both gold and silver, with the latter exhibiting better values of CD. The use of gold nanorods enables extending the response to the NIR from 800 to 1600 nm. For a fixed geometry engraved in the PDMS stamp, the choice of colloid (material, size and morphology) enables easily tuning the optical response of the plasmonic array.

Next, we investigate the angular dispersion from the triskelion array composed of 27 nm Ag colloids. In this case, an index matching oil with refractive index similar to the glass substrate was employed to study the 2D array in an homogeneous dielectric environment where the diffractive contributions of the lattice to the overall response can be clearly distinguished. The use of the index matching oil superstrate in the structures produces a narrowing of the lattice resonance and a red shift, but it decreases the $g$-factor under normal incidence for both colloids (Supplementary Fig. 26), which is attribute to the elimination of the asymmetry in the perpendicular direction provided by the substrate[45].

Figure 4a shows the angle-resolved transmittance for the Ag triskelion array under right- and left-circularly polarized light and linearly s-polarized (TE) illumination for angles of incidence (AOI) from −45° to 45°, in the Γ-M direction. For s-polarized light, the angular dispersion plots, $E$ (eV) versus parallel wavevector $k_{||}$ (nm$^{-1}$), clearly show the diffraction orders (1,1) and (−1,−1) of the hexagonal lattice at energies below 1.8 eV, for negative and positive AOI, repectively[49]. These diffracted orders are hybridized with the LSPR of the nanoparticle cluster in each unit cell, resulting in the SLRs with broader linewidths close to the LSPR energy. Under RCP and LCP illumination (Fig. 4a) the dispersion behavior of these SLRs modes is no longer symmetrical, in good agreement with previous observations[50]. A distinct preferential coupling of the LSPR with the diffraction mode is observed depending on the polarization state of the incident light, specifically for RCP light impinging at negative AOI. At low energies, the plasmonic mode couples with the (1,1) diffraction mode and a SLR is clearly observed. The inverse behavior is observed for left-circularly polarized light. For a comprehensive characterization of the angle-resolved chiral properties, taking into account nanoparticle sizes, azimuthal orientations, enantiomorphs and incidence directions, see Supplementary Figs. 27–33.

The different dispersion relation found at oblique incidence for RCP and LCP polarizations impinging on the index matched arrays is further manifested in the extracted $g$-factor, which is represented in Fig. 4b. Similar to the observations in split ring resonator arrays[50], the contribution to the $g$-factor from the 2D hexagonal lattice can be clearly identified. This type of angle-dependent contribution to the

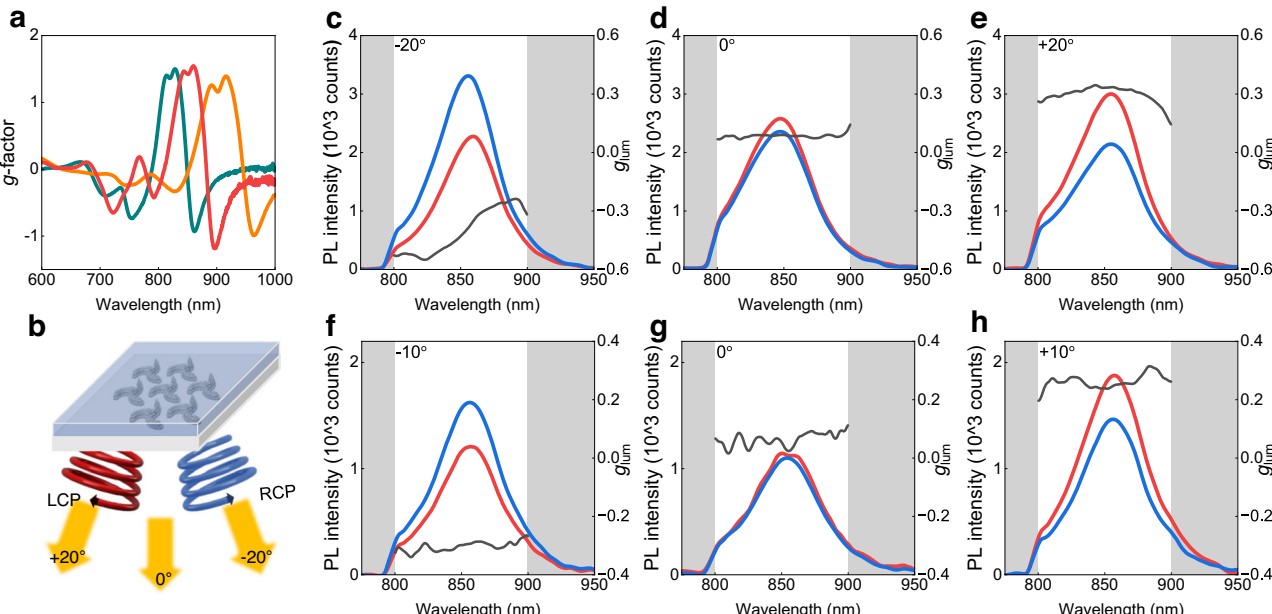

**Fig. 5 | CPL measured in Ag and Au triskelion arrays coated with IR-140 doped resist. a** Maximum $g$-factor (maximum differences in the extinction for left and right- circularly polarized light) measured from colloidal Ag arrays made with colloids of 27 (green line), 32 (red line) and 36 nm (orange line), covered with a thin layer of SU8 resist ($n \approx 1.6$) and collected at 21°, 31° and 42° angle of incidence. **b** Scheme of the resist coated nanoparticle arrays illustrating the different angles of collection at which the PL was collected. **c–h** Raw PL spectra measured under 760 nm excitation, represented by red and blue curves for left- and right-circularly polarized emission, respectively, and the corresponding photoluminescence dissymmetry factor $g_{lum}$ quantifying differences in emission between the two polarizations (gray lines). PL and $g_{lum}$ were collected from 36 nm Ag triskelion arrays at angles of −20° (**c**), 0° (**d**), and +20° (**e**), and from 54 nm Au triskelion arrays at angles of −10° (**f**), 0° (**g**), and +10° (**h**). Source data are provided as a Source Data file.

dichroism from the sample is found in different kinds of ordered arrays and is referred to as extrinsic chirality. Unlike intrinsic chirality which is independent of the angle of observation, the extrinsic contribution of the array changes signs when inspected from reversed angles of incidence. However, said contribution is typically very low for classical lattices, as confirmed by the hexagonal arrays of cylindrical nanoparticle clusters that we prepared using the same process ($g$-factors of ± 0.04 as seen in Supplementary Fig. 28). In the case of arrays of chiral objects, the extrinsic contribution of the lattice can really boost the $g$-factors of our array up to ±1.27 at a wavelength of 738 nm under observation at ± 14° AOI due to differential coupling to (1,1) and (−1,−1) diffraction modes (Fig. 4c). This differential coupling is also hinted for energies above 1.8 eV for ( ±1,0) and (0, ± 1) diffraction modes. Similar results were obtained for gold colloids, reaching $g$-factors up to 1 at 900 nm under 30° angle of observation (Supplementary Fig. 27). These triskelion 2D arrays offer high $g$-factor values that can be tuned with the AOI, thanks to the handedness-dependent excitation of plasmonic modes sustained by the triskelion 2D array. Furthermore, the $g$-factor in the NIR spectral range (750–950 nm) for incident angles ranging from 12° to 45° consistently exceeds 0.6 (Fig. 4d) and approaches 0.9 for angles around 25° and over a spectral range of ≈200 nm with a full width at half maximum (FWHM) for the resonance of less than 15 nm. This property could be exploited for dynamically controlled polarization spectral filters and optical rotators.

**Circularly polarized emission from dyes on 2D chiral arrays**
Finally, we illustrate how the strong dichroism of these colloidal chiral arrays can be harnessed, for instance, to produce CPL from achiral light sources. To this end, Ag or Au triskelion arrays were coated with a thin layer of dye-doped photoresist, i.e., SU8 (2000.5) doped with rhodamine B (RhB, Supplementary Figs. 34, 35) and IR emitting dye (IR-140, Fig. 5c–h), to probe two different spectral bands of the response of the plasmonic array. For details on the optical CPL setup see Supplementary Fig. 18. The CPL of the dyes layers coated onto the chiral

arrays was analyzed using figures of merit of the such as the PL dissymmetry factor ($g_{lum}$)

$$g_{lum} = 2\frac{(I_{LCP} - I_{RCP})}{I_{LCP} + I_{RCP}} \tag{2}$$

obtained from the difference between the intensities of the two polarizations where $I_{LCP}$ and $I_{RCP}$ are the intensities of the left and right circularly polarized components of the PL, respectively. It is worth mentioning that endowing chirality from organic dyes while maintaining brightness is a challenging task, $g_{lum}$ values are typically below $10^{-2}$[51]. Here we demonstrate orders of magnitude higher values for dyes operating in different parts of the spectral region by just placing them on top of the chiral array.

Figure 5a illustrates how the chiroptical signal ($g$-factor) of the Ag colloid arrays is influenced by the coating of a 250 nm thick SU8 layer (the matrix resist that hosts the dye) when measured at oblique incidence. This resist has a slightly higher refractive index ($n_{SU8} = 1.6$) than the index matching oil used before ($n_{IM} = 1.518$), which translates into a boost in the chiroptical response of the arrays, reaching values of $g$-factor of 1.5 when observed at 21° angle in the NIR region (800-900 nm).

To benefit from this strong chiroptical response, we mixed the SU8 resist with IR-140 fluorophore and used a supercontinuum laser centered at 760 nm for the optical excitation. The CPL from the dye-doped arrays is presented for both Ag (Fig. 5c-e) and Au (Fig. 5f-h). The CPL correlates well with the angular dispersion trends observed in Fig. 4b. As mentioned earlier, at normal incidence, the chiral response is reduced due to the effect of the superstrate. However, under oblique incidence, the difference between the intensity of emitted light in LCP (red line) and RCP (blue line) is notable, reaching $g_{lum}$ values exceeding -0.5 observed at −20° incidence (Fig. 5c) and the corresponding opposite values when observed at the positive angles (a $g_{lum} = 0.4$ at 20°) (Fig. 5e). Reassuringly, the $g_{lum}$ measured in the region out of the

pattern is 0 both at normal and oblique incidence (Supplementary Fig. 36), clearly pointing at the 2D array as the origin of the enhanced circular emission. For further studies on the CPL measured in these nanoparticle arrays such as fine-spectral tuning of the $g$-factor with the AOI, see Supplementary Fig. 37. In summary, architectures composed of nanoparticles arranged in hexagonal triskelion arrays can induce the chiral emission from non-chiral dyes and this chiroptical response can be actively tuned by simply modifying the angle of observation. A precise control can be achieved not only over the favored handedness but also precise tuning of the spectral position of the chiral emission maximum. This strategy enables producing high values of CPL while maintaining high brightness values from the dye (as supported by the raw PL data presented here).

A similar study was performed by mixing RhB with the resist and optically excited using a 532 nm laser, showing the characteristic PL signal of the dye in the 600–700 nm window (see Supplementary Fig. 34). The CPL values obtained therein for the non-chiral dye are lower, since they do not benefit from the intense chiral lattice resonances exhibited in the 800–900 nm region. Still, exploiting the higher orders of diffraction supported by the array allowed us to obtain $g_{lum}$ values ranging from $\pm 0.2$ to $\pm 0.4$ depending on the handedness of the triskelia used. Moreover, we observe a dependence on both the sign and magnitude of the $g_{lum}$, which is contingent upon the direction of the incident light (see Supplementary Figs. 34, 35)[52].

## Discussion

We have shown the scalable fabrication of nanoparticle chiral arrays with the aid of pre-patterned elastomeric stamps and silver and gold colloids. The as-prepared 2D chiral plasmonic arrays display strong optical activity reaching $g$-factors of 0.4 in the case of Ag colloids (0.18 in the case of Au) under normal incidence. However, if the structure exploits also the extrinsic chirality contribution of the chiral lattice by the addition of a superstrate, even larger values of $g$-factor can be attained reaching $g$-factors of 1.5 at 830 nm at 21° AOI for Ag colloids ($g$-factor = 1 for Au colloids). We have also illustrated how to harness the chiroptical response of the 2D arrays by depositing a thin layer of dye-doped photoresist. Coupled to the metasurface, the fluorophore showed circularly polarized luminescence with a dissymmetry factor ($g_{lum}$) up to 0.55 tunable with the AOI. In sum, we have demonstrated the potential of templated assembly for the production of tunable chiral metasurfaces from conventional colloids that can lead to the incorporation of chiral nanostructures in a wide variety of applications.

## Methods

### Materials and apparatus

Hydrogen tetrachloroaurate trihydrate ($HAuCl_4 \cdot 3H_2O$, $\geq 99.9\%$), hexadecyltrimethylammonium chloride (CTAC, 25 wt.% in water), hexadecyltrimethylammonium bromide (CTAB, Lot 10211934, 98%), sodium citrate tribasic dihydrate (SC, $\geq 99.9\%$), tannic acid (TA, $\geq 99.9\%$), poly(ethylene glycol) methyl ether thiol (PEG-SH, $M_w$ 2000 g mol$^{-1}$), trichloro (1H,1H,2H,2H-perfluorooctyl) silane (97%), 5-bromosalicylic acid ( > 98.0%), silver nitrate ($AgNO_3$, $\geq 99\%$), Rhodamine (RhB, $\geq 99.9\%$) and IR emitting dye 5,5'-dichlor-11-diphenylamin-3,3'-diethyl-10,12-ethylenthiatricarbocyaninperchlorat (IR-140, 95%) were purchased from Sigma-Aldrich. Sodium borohydride ($NaBH_4$, 99%) and sodium hypochlorite (NaClO, 10–15 wt.% active chlorine) were purchased from Acros Organics. L-ascorbic acid (AA, $\geq 99\%$) was purchased from Alfa Aesar. Polydimethylsiloxane (PDMS, Sylgard 184) was purchased from Dow Corning (Michigan, USA). OrmoStamp (Lot 37416) was purchased from Micro Resist Technology. 7.0-8.0 wt.% vinyl methyl siloxane, 1,3,5,7-tetraacetylcyclosilane, platinum catalyst and 35% hydroxysiloxane were purchased from Gelest (USA). Epoxy photoresist SU8 2000.5 (14 wt.%) was purchased from Kayaku Advanced Materials (Japan). Index matching oil ($n = 1.518$) was purchased from Olympus (Janpan). Acetone

($\geq 99.6\%$), isopropanol (IPA, $\geq 99.8\%$) and ethanol ($\geq 96\%$) were purchased from Labbox. All chemicals were used as received.

Each glass substrate (Labbox Spain, $20 \times 20$ and $24 \times 24$ mm$^2$) used was cleaned in the ultrasonic bath 10 min with acetone, IPA and water respectively. The glassware used in the synthesis of colloids was washed in aqua regia (3:1 HCl:HNO$_3$) and rinsed with water, and dried before use.

Absorbance spectra were recorded by Hitachi U-3000 spectrophotometer. TEM images were recorded by JEOL 1210. SEM images were recorded by FEI QUANTA 200 Field Emission Gun and FEI Magellan 400 L microscopes. CD optical measurements were recorded by a custom-built optical setup (Supplementary Fig. 17). All optical components were automatically controlled by software LabView NXG to ensure repeatability of measurements.

### Triskelion pattern design

Nanoimprinting stamp masters were fabricated on silicon wafers using electron beam lithography (ConScience AB) to create a 200-nm deep 2D hexagonal lattice with triskelion-shaped unit cells, (see Supplementary Fig. 1). Two $4 \times 4$ mm$^2$ areas were patterned with L- and R-triskelion lattices. FDTD (Lumerical by Ansys) simulations optimized the triskelion design and orientation on a hexagonal lattice to maximize the $g$-factor, using a 100-nm thick Ag layer on glass substrates to evaluate CD and LD. The hexagonal lattice was chosen over a square lattice due to its threefold symmetry and superior simulated CD and LD performance.

### Fabrication of the PDMS molds

The engraved silicon wafers were silanized with 8 μL of trichloro (1H,1H,2H,2H-perfluorooctyl) silane in a vacuum for 30 min to prevent adhesion during replication. Intermediate working masters were fabricated using UV nanoimprinting. Ormostamp resist was applied to the silanized master at 60 °C, covered with a glass substrate to prevent bubbles, and cured under UV light for 10 min. The working master was separated by heating from 110 to 180 °C to exploit thermal expansion differences and then silanized like the original master.

Next, hard-PDMS (h-PDMS) stamps were prepared by mixing vinyl-methyl-siloxane, 1,3,5,7 tetraacetyl cyclosilane, platinum catalyst, hydroxysiloxane and toluene. The working master was covered with the mixture and left for 1 h to let toluene evaporate. Next, the wafer was cured at 60 °C for 1 h. Soft PDMS was prepared by mixing monomer and curing agent (10:1 ratio), degassed in vacuum for 2 h, and poured over the cured h-PDMS layer to provide back support. After a 30-min vacuum treatment, the assembly was cured at 100 °C for 2 h. The molds were then separated and cut to ideal size.

### Synthesis of silver nanospheres

Synthesis of silver nanospheres was based on the work of Bastús et al. [53]. To synthesize 20 nm silver nanospheres, 200 mL of 5 mM SC solution was boiled, and 8 mL of 2.5 mM TA solution and 2 mL of 25 mM AgNO$_3$ solution were added at 1-min intervals while stirring for 10 min. The nanospheres were collected by centrifugation at $16,099 \times g$ for 20 min. Larger silver nanospheres were grown using 20 nm Ag seeds solution without centrifugation. After the above initial reaction, the seeds solution was cooled to 90 °C, and different volumes of SC, TA and AgNO$_3$ solution were added to solution to obtain the desired final silver size, as detailed in Supplementary Table 1.

### Synthesis of gold nanospheres

Synthesis of gold nanospheres was based on the work of Hanske et al. [54]. First, 5 mL of 0.1 M CTAC solution was mixed with 50 μL of 50 mM HAuCl$_4$ solution, followed by 200 μL of 20 mM fresh NaBH$_4$ solution. After 3 min of stirring, the mixture was diluted 10 times with 0.1 M CTAC solution. Under stirring, 10 mL of 25 mM

CTAC solution was added to 900 µL of the diluted seed solution, followed by 40 µL of 100 mM AA solution and 50 µL of 50 mM HAuCl₄ solution. The mixture was left undisturbed at room temperature for at least 10 min. The resulting 10 nm gold nanospheres solution can be stored at room temperature for long-term use. For seeded growth of different sizes, 100 mL of 25 mM CTAC solution, 250 µL of 0.1 M AA solution and 250 µL of 50 mM HAuCl₄ solution were used in each reaction for different gold nanoparticles size in this step, but volumes of 10 nm gold seeds solution were changed, as detailed in Supplementary Table 2. To further optimize the shape of the synthesized gold nanospheres for an ideal spherical form, above ideal size of gold nanoparticles solution was heated to 30-35 °C, and different volumes of 1–1.5 wt.% NaClO solution and 50 mM HAuCl₄ solution were added, as detailed in Supplementary Table 2.

### Synthesis of gold nanorods

Synthesis of gold nanorods was based on the work of Ye et al.[55]. 4.7 mL of 0.1 M CTAB solution was mixed with 25 µL of 50 mM HAuCl₄ solution under stirring, followed by 0.3 mL of 0.01 M fresh NaBH₄ solution. After 2 min of stirring, the mixture was left undisturbed for at least 30 min.

100 mL of 0.1 M CTAB solution was mixed with 200 mg of 5-bromosalicylic acid and stirred at 60 °C until dissolved, then cooled to 30 °C. Next, 1.92 mL of 0.01 M AgNO₃ solution was added and stirred for 15 min, followed by 2 mL of 50 mM HAuCl₄ solution and stirring for 15 min. Finally, 540 µL of 0.1 M AA solution was added and stirred for 30 s, turning the solution colorless.

160 µL of the seed solution was added to the growth solution, stirred for 1 min, and left for 12 h at 30 °C. The resulting gold nanorods were centrifuged at $11,180 \times g$ to remove unreacted chemicals, and water was added to replace CTAB for the PEG-SH connection in the next step.

### Surface functionalization

Surfaces were functionalized with PEG-SH ligands by adding the 1 mg mL⁻¹ PEG-SH solution to colloidal solutions, using $H_2O$ or 300 µM CTAC as solvents. The exchange process lasted 12 h, followed by three centrifugation steps to remove unreacted PEG-SH, and the colloids were stored in minimal volume.

### Colloidal dispersion concentration calculation

The concentration of Ag nanoparticles was estimated from the absorbance value at the LSPR maximum ($\lambda_{max}$). According to the extinction coefficients ($\varepsilon$) calculated by Paramelle et al.[56]. (Supplementary Table 3), the corresponding concentration of Ag nanoparticles was obtained by the Beer-Lambert law as $c = A/(\varepsilon b)$, where $b = 1$ cm is the optical path, $A$ the absorbance and $c$ is the NPs concentration. To calculate the molar concentration of $Ag^0$, we took into account that the mass density of silver is 10.49 g cm⁻³, the molar mass of silver is 107.87 g mol⁻¹, and Avogadro's number is $6.022 \times 10^{23}$. The relevant calculation procedure can be referred to the Supplementary Eqs. (1)–(3).

A different approach was used to calculate the gold concentration, the absorbance of the gold colloids at 400 nm ($A_{400}$) was measured. As described by Scarabelli et al.[57], there is a linear relationship between $A_{400}$ and $Au^0$, during $Au^{0r}$ concentration below 0.8 mM, and $A_{400} = 1.2$ when $[Au^0] = 0.5$ mM. The concentration of $Au^0$ in the solution could be calculated. The relevant calculation procedure can be referred to the Supplementary Eq. (4).

### Templated assembly of the nanoparticle-based chiral arrays

A volume of 1.5 µL of gold nanospheres/nanorods dispersion was placed on the glass substrate and immediately covered with the PDMS stamp and left overnight. Next day, the stamp was removed leaving a nanoparticle assembly on the glass.

### Preparation of index match layer and SU8 resist coating

Index matching oil ($n = 1.518$) used for the experiments shown in Fig. 4: A drop of oil was added to the sample and immediately covered with a $24 \times 24$ mm² coverslip glass. Resist layer used in experiments shown in Fig. 5: The epoxy photoresist SU8 2000.5 was used here to host an organic dye and as a coating to the samples leading to an increase in the g-factor of triskelion arrays. The SU8 resist was diluted to 7 wt.% using thinner and spin coated at 2000 rpm, and UV cured for 5 min and consolidated at 90 °C for 5 min. Dye doping, RhB or IR-140 (Fig. 5), was included in the 7% SU8 mixture and deposited on the samples by spin coating following the same procedure.

## Data availability

The data that support the findings of this study are available from Zenodo[58] and from the corresponding authors upon request. Source data are provided with this paper.

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

## Acknowledgements

X. Q. and J.M.C. kindly acknowledge the auspices of the Ph.D. program in Materials Science from Universitat Autònoma de Barcelona. The authors are grateful to Dr. Bernat Mundet for his valuable assistance with EELS characterization. Authors acknowledge the use of instrumentation financed through Grant IU16-014206 (METCAM-FIB) to ICN2 funded by the European Union through the European Regional Development Fund (ERDF), with the support of the Ministry of Research and Universities, Generalitat de Catalunya. This work has received funding from the Spanish Agencia Estatal de Investigación (MCIN/AEI/10.13039/501100011033) through grants PDC2021-121475-I00, TED2021-132807B-I00, PID2022-141956NB-I00 (OUTLIGHT), and CCEX2023-001263-S (Spanish Severo Ochoa Center of Excellence program) and from the Generalitat de Catalunya (2021-SGR-00444). This research was also supported by the EIC PATHFINDER OPEN project 101046489 (DYNAMO) and EIC PATHFINDER CHALLENGES project 101162112 (RADIANT), funded by the European Union. X.Q. is grateful to the China Scholarship Council for funding (CSC No. 202206790019).

## Author contributions

X.Q., L.A.P., and A.M. conceived the project. X.Q. fabricated the samples. X.Q., L.A.P., and J.M.C. performed the optical characterization and analyzed the data. L.A.P. performed the FDTD simulations. M.G. and M.I.A. designed the chiral metasurface. L.A.P. and A.M. supervised the project. All the authors contributed to the manuscript.

## Competing interests

The authors declare no competing interests.
