## [Transparent Peer Review File · Nature Communications]

Chiral plasmonic superlattices from template-assisted assembly of achiral nanoparticles

Corresponding Author: Dr Agustín Mihi

Version 0:

Reviewer comments:

Reviewer #1

(Remarks to the Author)

It is a beautiful work. This work demonstrates a lot of new possibilities to achieve a high g factor by designing chiral patterns. There are several questions that need to be addressed.

1. Compared to the gammadion type or other chiral lattice, why does this structure of nanoparticles' assembly exhibit higher g factor? Do you expect similar phenomenon if we make the same pattern with the conventional lithography? In your simulation, the author only considered the pattern without considering the effect of assembly of nanoparticles.
2. The reviewer want to know the data with the p polarization. Typically, p-polarization is more effective to maximize the lattice mode.
3. "In contrast, at positive angles, the plasmonic mode 238 does not couple with the (-1,-1) diffraction mode; instead, it couples with the (0,-1) and (-239 1,0) modes." This discussion seems to strange. In Fig4 A, please specify the lattice mode.
4. In fig 4C, the origin of all the peaks should be assigned. Depending on the incident angle, the lattice mode will be shifted and the coupling behavior of LSPR from pattern and Lattice mode will be changed. The detailed coupling mode and mechanism need to be identified. This understanding will be useful for the future research
5. The reviewer expect that higher incident angle with p-polarization can also exhibit high g factor close to 1.0.

Reviewer #2

(Remarks to the Author)

The authors demonstrate the fabrication of chiral metasurfaces using a combination of soft lithography techniques and plasmonic nanoparticles. Specifically, solutions of Ag or Au nanostructures are used as an "ink" which are then patterned by a PDMS stamp into a hexagonal array of chiral triskelion-shaped structures composed of closely-packed nanoparticles. The method allows for large areas of well-defined chiral objects which show superior chiroptical response (as determined by g-factor) in comparison to triskelia composed of bulk, rather than nanoparticulate, metals. Extensive optical characterization including band diagrams and EELS measurements provide convincing evidence of extremely strong chiroptical dissymmetry. Finally, photoluminescent Rhodamine compounds can be spin coated on top of the particle-based triskelion array to produce a metamaterial with circularly polarized photoluminescence.

Overall I found the paper to be well-written, the data to be extremely convincing as it combines theory and multiple independent measurement techniques, and the results to be exciting. However, I cannot recommend publication because I believe the novelty of the work to be incommensurate with the impact of a journal like Nature Communications. Although the characterization is extremely thorough and impressive, very little of it is surprising given the previous work in this field. For example, a cursory search uncovered a 2021 paper in which a similar soft lithography technique was used to pattern spherical Au nanoparticles into an array of chiral structures (see 10.1021/acsmaterialslett.0c00535). Most of the trends in

optical properties are expected, as evidenced by the excellent agreement with FDTD simulations. Indeed the authors appear to have achieved what are possibly record-breaking g-factor values for chiral plasmonic structures but this appears to be rather more of an (admitted beautiful and high-quality) extension of previous work as opposed to something fundamentally new and unexpected. I therefore believe the manuscript to be more appropriate for a more specialized journal. Below are a few minor points that had no impact on my opinion:

- In the SI under the “synthesis of gold nanorods” section, CTAB is described as a solvent which is incorrect as it is a solid at the temperatures described. I’m fairly certain it is being used in an aqueous solution, which should be specified.
- Table S2 describes hypochloric acid as HClO but should be HClO. In addition the methods describe using sodium hypochlorite which is different

Reviewer #3

(Remarks to the Author)

In this manuscript Mihi and co-workers create chiral plasmonic lattices from the ‘most common metal colloids’. Plasmonic structures offer a new approach to boost chiral light-matter interactions and achieve strong dissymmetry factors, but sometimes suffer from complicated fabrication requirements and have a limited spectral range. Metal colloids offer the potential to operate in the visible regime, with scalable fabrication processes and often strong chiroptical phenomena. Hexagonal arrays of metal colloids can achieve strong g-factors under normal incidence, and even stronger from different viewing angles. They go on to show that placing an achiral NIR dye on top of the plasmonic array can trigger CPL emission from the dye, where the collection angle impacts the sign/magnitude of the chiroptical response. The article is interesting, well-written and thorough. I am not sure it is appropriate for the broad readership of Nature Communications – similar work has ended up in Scientific Reports – but the science is sound. A few small points below.

Figure 2 in the SI is difficult to read and could do with a legend (e.g. saying CD: black line, LD: red line, transmittance: dashed blue line). What happens to the optical response if you stack two layers of triskelions?

Why do the left- and right-handed systems not have equal and opposite optical responses (Figure 3B, D and F)?

Why do the Ag nanospheres generate stronger chiroptical responses than the Au nanospheres? Can the authors comment on why the line shape is so different for the two metals (Ag, 3B, seems to be dominated by absorption of one handedness, whilst Au, 3D, looks more balanced)?

The authors should include a photograph/ diagram showing how the angle resolved characterisation (Fig 4, SI Section S9) were performed, as it would be clearer what a ‘negative AOI’ etc is.

The authors should label the y-axis on Figure 5A g-PL, as they have previously used g-factor for g-abs. The angular resolved g-factor spectra are quite confusing, particularly how there is strong g-PL even when there is negligible difference in the red and blue curves (e.g. Fig 5C-E, Fig5F-H). Have the authors measured these spectra more than once, are these results/calculations reproducible?

The g-factor spectra (e.g. Figure 3D and 3F) are very noisy despite the strong dissymmetric response. Can the authors comment on why?

The SI contains a lot of data which are barely discussed. For example, the main manuscript mentions a “sign-flip of the g-factor”, but it is not clear in the text.

Given the strongly angle-dependent response, how do the authors propose this can be used in a technological context?

The references seem to omit the work of nearby researchers, e.g. <https://www.nature.com/articles/s41598-021-03908-2>.

In conclusion, this is an interesting paper which uses a simple, scalable strategy to achieve strong g-factors. I am still not convinced it will appeal to the multi-disciplinary readership of Nature Communications, nor how translatable the approach is to technological contexts/device architectures.

Reviewer #4

(Remarks to the Author)

Qi et al. introduce a process to fabricate complex, chiral plasmonic lattices from simple building blocks. The authors use nanoimprint lithography to create chiral templates in a PDMS film, into which they accurately deposit simple, spherical (and rod-shaped) nanoparticles. As a result, complex, chiral lattices arise from simple building blocks. The authors explore their chiroplasmonic properties, show that the lattice resonances can be tailored via the particle size and demonstrate high chiral signals at the spectral positions of these resonances. Finally, they demonstrate that the chiral signature of the created metasurfaces can be used to manipulate light emission from fluorescent dyes in proximity. The manuscript demonstrates a simple approach to highly complex, functional metasurfaces with an attractive combination of a scalable process, tailored chiroptical properties and an interesting application in light emission manipulation. I believe that the manuscript can be of interest to the broad readership of nature communications with some additional clarifications and revisions, as detailed below.

Introduction:

the overview of literature reports could be a bit more detailed to outline the field. The authors mention some examples of top-down (nanofabrication) strategies (e.g. Refs.11-12), and chiral, dispersed colloids (e.g. Refs 27-28) to make their case that simpler strategies are useful (to which I agree). To provide a more complete picture of the field, the authors should consider adding some seminal reports for top-down lithographic approaches – which essentially paved the way for further developments of simpler strategies, such as Liu et al., Nature Photonics 2009, 3, 157 (which paved the way for the field), or Kim et al., Nature 2022, 612, 470 (which developed sensors based on such resonances). In addition, there are several directions in the direction of simpler approaches using self-assembly that may be worth discussing to set the context for their

manuscript, including colloidal lithography (Goerlitzer et al., *Adv. Mater* 2020, 32, 2001330), induced chirality from achiral plasmonic materials (e.g. Maoz et al., *Nano Lett.* 2013, 13, 1203, Goerlitzer et al., *ACS Photonics* 2023, 10, 1821), or the stacking of achiral objects into 3D chiral structures (e.g. Probst et al, *Nature Materials* 2021, 20, 1024).

The authors may also want to give some credit to previous work on nanoimprint lithography, which they use in their manuscript without much detail on the fabrication process. Briefly introducing the technique with references would help a reader get access to the framework and set the work in context of previous approaches; including work of the authors themselves (e.g., Kraus et al., *Nature Nanotech.* 2007, 2, 570; Scarabelli et al., *Acc. Mater. Res.* 2021, 2, 9, 816; Hanske et al., *Nano Lett.* 2014, 14, 12, 6863, Matricardi et al. *ACS Nano* 2018, 12, 8, 8531)

Structure Fabrication:

Why is the g-factor enhanced in the simulations of nanoparticle-based triskelia as compared to bulk gold ones (Figure 1C, D)?

What is the impact of structural imperfections, such as missing arms of the triskelion, particles deposited aside, missing particles, or polydispersity? From the SEM images, the fabrication seems to be of very high quality, but such effects will be found in such fabrications in some areas – especially when considering scaled fabrication.

Optical characterisation

In Figure 4, the authors show the spectral properties of triskelia arrays made of different particle building blocks. They assign the peaks between 600-800nm to the collective lattice resonance, and the higher energy features (400-600nm) to the LSPRs. If the collective resonance is caused by the triskelia themselves (which are formed by many particles in close proximity), why do they shift with the primary particle size? Would one not expect the local mode to change, while the array properties remain similar as the individual triskelia act as a single entity – or is this picture too simplified?

The fluorescence of the pure Rhodamine B dye is in the range of 600nm-700nm (as the authors state in their manuscript).

Why is the Circularly polarized photoluminescence measured when the dye was added as a superstrate to the triskelia arrays measured between 800nm-900nm (Figure 5)?

In Figure 5, the authors show the chiral properties (g-factor) of three different Ag NP samples, but in C-E and F-E they compare silver and gold particles with different sizes? Would it not be clearer to show the spectra of the arrays used for the PL studies in Figure 5a to allow a clear correlation between resonance position and PL enhancement?

Version 1:

Reviewer comments:

Reviewer #1

(Remarks to the Author)

Recommendation: Accept without further revision

The author responded all our queries and all the reply from the author is satisfactory. Therefore, I recommend the manuscript to publish in its current form.

Reviewer #3

(Remarks to the Author)

The authors have provided adequate responses to the Reviewer's queries. While they have clarified some sections, they have not taken on board all the changes. For example, in Figure 5, they use the labels "g-factor" and "g-lum". Both are dissymmetry (g-factors). They should indicate that one is for absorption and the other for luminescence on the legend, and the excitation wavelength for luminescence should be included in the caption.

The new diagram in Figure S17 is useful to explain the AOI, and much appreciated by the reviewer.

The authors have included some 'chiral photodetectors', where they coat conventional silicon photodetectors with the metasurfaces, in an attempt to show technological applications. These data do not make it into the manuscript/SI, and I am a little confused. What is the coverage of these devices? From the accompanying photograph, it looks relatively poor (<40%?). It would be good to put the accompanying g-abs so we can track whether the peaks in g-ph correspond to peaks in absorption dissymmetry. Where is the laser spot focussed on the photodetectors?

In summary, while the authors have boosted the g-factors of chiral metasurfaces, the paper still does not contain novel scientific developments/technological demonstrations, and is more appropriate for another, more specialized, journal.

Reviewer #4

(Remarks to the Author)

The authors have adequately clarified my remarks on the initial submission. I believe that the combination of a comparably simple fabrication process (soft lithography using simple building blocks vs. synthesis of complex-shaped particles) with a surprisingly strong performance (i.e. better than bulk metal films) and a thorough characterization of structure and optical properties makes this manuscript valuable to a broad scientific audience – as targeted by nature communications.

As a minor remark, I believe that the answer the authors provided to my remark on the influence of structural imperfections (which will increase extinction without contributing to chiral signals) may be mentioned in the main text – simply to describe

a potential disadvantage of the system along all the advantages the authors already introduce.

Version 2:

Reviewer comments:

Reviewer #3

(Remarks to the Author)

The authors have provided a satisfactory response to the comments of all reviewers, and the paper can be published in its current and corrected form.

RESPONSE TO REVIEWERS

Reviewer #1

It is a beautiful work. This work demonstrates a lot of new possibilities to achieve a high g factor by designing chiral patterns. There are several questions that need to be addressed.

Authors: We are very grateful to the reviewer for the kind words towards our work.

1. Compared to the gammadion type or other chiral lattice, why does this structure of nanoparticles' assembly exhibit higher g factor? Do you expect similar phenomenon if we make the same pattern with the conventional lithography? In your simulation, the author only considered the pattern without considering the effect of assembly of nanoparticles.

Authors: There are several questions here that we will attempt to address separately in the following lines:

Compared to the gammadion type or other chiral lattice, why does this structure of nanoparticles' assembly exhibit higher g factor?

Authors: This is a very relevant question that points towards the importance of the proper design of the chiral geometry used. As expected, not all chiral shapes exhibit the same values of circular dichroism (CD) and not all of them operate in the spectral range of interest. To arrive to our design of the triskelion array, we performed several FDTD based simulations of the triskelion array (changing dimensions, rotation angle respect to the lattice director vectors, periodicity, etc.) looking for the geometry that maximized the values of CD in the visible-near infrared (Vis-NIR). We found the triskelion array configuration presented in the paper outperformed other geometries. In order to illustrate this point, we have fabricated different chiral arrays using the same method and the same colloid type (30-50 nm Au colloids) and we have compared their CD performance. A new section (section 7) has been added to the SI showing this comparison:

Section 7 of the SI reads: Figure S16 presents a comparative analysis of the chiral properties of various architectures constructed from metallic colloids (30-50 nm gold nanospheres). In the SEM images it can be observed that template-directed self-assembly facilitated the fabrication of highly ordered lattices with different chiral geometry unit cells. The geometries include G-shaped structures (from the literature, *N. Chiang et al. ACS Materials Lett. 2021, 3, 3, 282–289*), tilted linear segment arrays, gammadions, and triskelions (all prepared in our lab). The G-shaped structures exhibit the largest lattice parameter (12 μm), the remaining three have lattice parameters of 500, 600, and 500 nm, respectively.

CD spectra in the Vis-NIR region demonstrate a pronounced dependence on the specific geometry. G-shaped structures exhibit broad CD bands with peak intensities around 40 mdeg. In contrast, segment line arrays display significantly higher maximum CD signals, exceeding 400 mdeg. Both gammadions and triskelions exhibit the highest CD responses with sharp, intense bands. The maximum CD signal for gammadions is centered near 900 nm in the NIR, whereas triskelions show the highest CD signal blue-shifted towards the visible region.

Figure S16. Scanning electron microscopy (SEM) images and circular dichroism (CD) spectra of various plasmonic architectures constructed using spherical gold colloids. G-shape values are extracted from *N. Chiang et al. ACS Materials Lett. 2021, 3, 3, 282–289* and the rest were prepared for this response. Reproduced with permission from Large-scale soft-lithographic patterning of plasmonic nanoparticles. Chiang, N., Scarabelli, L., Vinnacombe-Willson, G. A., Pérez, L. A., Dore, C., Mihi, A., Jonas, S. J., Weiss, P. S. *ACS Mater. Lett.* 3, 282-289 (2021). Copyright 2021 American Chemical Society.

Do you expect similar phenomenon if we make the same pattern with the conventional lithography?

Indeed, a chiral response is expected from chiral geometries fabricated with conventional lithographies, as has been done previously in the literature. In order to ascertain the influence of having the same geometry made of gold colloids versus solid continuous gold, we have performed FDTD simulations for our triskelion design (Fig R1). These calculations show how the circular dichroism appears in the same spectral range for both systems, but the values of g-factor are higher in the particulate case for the thicknesses and geometry considered herein.

Figure R1. Simulated optical response of a 2D hexagonal array of triskelia composed of a) 100 nm height Bulk gold (Johnson and Christy) and b) 27 nm gold nanoparticles hexagonally packed.

2. The reviewer want to know the data with the *p* polarization. Typically, *p*-polarization is more effective to maximize the lattice mode.

Authors: We have included the Transmittance spectra for the arrays under linearly *p*-polarized light in the SI Fig S31.

3. "In contrast, at positive angles, the plasmonic mode 238 does not couple with the (-1,-1) diffraction mode; instead, it couples with the (0,-1) and (-239 1,0) modes." This discussion seems to strange. In Fig4 A, please specify the lattice mode.

Authors: We have revised this paragraph and described better this section of the work.

"Fig. 4A shows the angle resolved transmittance for the Ag triskelion array under right- and left-circularly polarized light (left and right panels, respectively) and linearly *s*- polarized (TE) illumination (center panel) for angles of incidence (AOI) from -45° to 45° , in the Γ -M direction. For *s*-polarized light, the angular dispersion plots, E (eV) versus parallel wavevector k_{\parallel} (nm^{-1}), clearly show the diffraction orders (1,1) and (-1,-1) of the hexagonal lattice at energies below 1.8 eV, for negative and positive AOI, respectively.⁴⁹ These diffracted orders are hybridized with the LSPR of the nanoparticle cluster in each unit cell, resulting in the SLRs with broader linewidths close to the LSPR energy. Under RCP and LCP illumination (Fig. 4A) the dispersion behavior of these SLRs modes is no longer symmetrical, in good agreement with previous observations.⁵⁰ A distinct preferential coupling of the LSPR with the diffraction mode is observed depending on the polarization state of the incident light, specifically for RCP light impinging at negative AOI. At low energies, the plasmonic mode couples with the (1,1) diffraction mode and a SLR is clearly observed. The inverse behavior is observed for left-circularly polarized light."

4. In fig 4C, the origin of all the peaks should be assigned. Depending on the incident angle, the lattice mode will be shifted and the coupling behavior of LSPR from pattern and Lattice mode will be changed. The detailed coupling mode and mechanism need to be identified. This understanding will be useful for the future research

Authors: In response to the reviewer's comment, we have carefully examined and revised the text associated with Figure 4C. We have identified the peaks corresponding to the various modes observed in the spectra. Furthermore, to provide a deeper understanding of the underlying physics, we have included a theoretical model in the Supporting Information section 2. This model visualizes the coupling mechanism between the chiral mode of each individual triskelion and the collective lattice mode, as depicted in Figure S3.

To elucidate the mechanism underlying the chiral behavior of triskelion arrays, we computationally modeled the optical properties of isolated triskelions and triskelion lattices. Figure S3A presents the extinction coefficients of a single triskelion under circularly polarized illumination. A pronounced extinction maximum is observed near 600 nm for both circular handedness, attributable to the localized surface plasmon resonance (LSPR) of the triskelion shape. This resonance arises from the collective response of its near-field coupled constituent nanoparticles. The observed differential optical response confirms the chiral nature of the triskelion. The g-factor, as depicted in Figure S3C, exhibits a chiral response profile within the 600-700 nm range. Furthermore, Figure S3B displays the transmission spectra of hexagonal triskelion arrays with a period of 500 nm for both circular polarization handedness. The dashed red vertical line indicates the Rayleigh Anomaly of the array (RA), at that wavelength a significant difference in transmittance between the two polarizations is observed. This finding is corroborated by the calculated g-factor for the arrays, which reveals an enhanced chiral response compared to an isolated triskelion (Figure S3C, red line). These results suggest that the hybridization of the chiral LSPR mode of each triskelion with the lattice diffraction mode, leads to a surface lattice resonance (SLR) with a strong chiral character, becoming the dominant mechanism responsible for the amplified chiral response in triskelion arrays.

Figure S3. FDTD simulation of A) the extinction coefficient of an isolated triskelion, B) transmittance spectra of triskelions arrays under circularly polarized light. C) Compared g-factor response of single triskelion (black) versus triskelion arrays (red). LP= 500 nm, Au NPs 27nm, interparticle distance 6 nm.

5. The reviewer expect that higher incident angle with p-polarization can also exhibit high g factor close to 1.0.

Authors: We appreciate the reviewer's comment. However, it should be noted the g-factor is intrinsically defined in terms of the interaction between a material and left and right- circularly polarized light. As a result, experimental determination of the g-factor necessitates the use of circularly polarized light. Consequently, it is not feasible to extract g-factor values from spectra acquired using p-polarized light, irrespective of the incident angle.

Incidentally, a comprehensive angular characterization of the optical response of the triskelion arrays can be found in the SI, Figures S27-S32.

Reviewer #2 (Remarks to the Author):

The authors demonstrate the fabrication of chiral metasurfaces using a combination of soft lithography techniques and plasmonic nanoparticles. Specifically, solutions of Ag or Au nanostructures are used as an “ink” which are then patterned by a PDMS stamp into a hexagonal array of chiral triskelion-shaped structures composed of closely-packed nanoparticles. The method allows for large areas of well-defined chiral objects which show superior chiroptical response (as determined by g-factor) in comparison to triskelia composed of bulk, rather than nanoparticulate, metals. Extensive optical characterization including band diagrams and EELS measurements provide convincing evidence of extremely strong chiroptical dissymmetry. Finally, photoluminescent Rhodamine compounds can be spin coated on top of the particle-based triskelion array to produce a metamaterial with circularly polarized photoluminescence.

Overall I found the paper to be well-written, the data to be extremely convincing as it combines theory and multiple independent measurement techniques, and the results to be exciting. However, I cannot recommend publication because I believe the novelty of the work to be incommensurate with the impact of a journal like Nature Communications. Although the characterization is extremely thorough and impressive, very little of it is surprising given the previous work in this field. For example, a cursory search uncovered a 2021 paper in which a similar soft lithography technique was used to pattern spherical Au nanoparticles into an array of chiral structures (see 10.1021/acsmaterialslett.0c00535). Most of the trends in optical properties are expected, as evidenced by the excellent agreement with FDTD simulations. Indeed the authors appear to have achieved what are possibly record-breaking g-factor values for chiral plasmonic structures but this appears to be rather more of an (admitted beautiful and high-quality) extension of previous work as opposed to something fundamentally new and unexpected. I therefore believe the manuscript to be more appropriate for a more specialized journal. Below are a few minor points that had no impact on my opinion:

Authors: We are thankful to the reviewer for recognizing the “record-breaking g-factor values” obtained and we would like to kindly explain in the following lines the novelty of our work.

First, we would like to describe the impact of our work within the current state of the art. The development of chiral metal colloids with **g-factors up to 0.2** (González-Rubio *et al.*, *Science* 368, 1472–1477 (2020), Lu *et al.*, *Science* 371, 1368–1374 (2021), Lee *et al.* *Nature* 360, 556 (2018)) indicates a clear intent to harness a strong chiroptical response and bring it closer to practical applications using scalable techniques. However, the production of chiral colloids requires extraordinary craftsmanship (so far mostly shown for Au), and it is worth mentioning that such strong g-factors are reported from solution and hardly ever measured on a substrate. This is explained in the work by Karst *et al.* (*ACS Nano* 13, 8659–8668 (2019)), where it is analyzed how the random orientation or aggregated state of the chiral particles decreases or even cancels out the chiral near fields responsible for the optical activity of the material. *In sum, it is very challenging to produce, in a scalable manner, chiral architectures on a substrate.*

Second, we would like to discuss previous attempts to use templates to achieve chiral architectures using non-chiral colloids. This method makes use of organic templates such as nanotubes or DNA, to produce a chiral assembly of chiral colloids. The most relevant works in the field (see table) have shown **g-factors around 0.12.**

The work mentioned by the reviewer (see 10.1021/acsmaterialslett.0c00535) describes a soft lithography technique used to pattern spherical Au nanoparticles into an array of 8 micron G-shapes and produced a **g-factor value of 0.002 at 900 nm** (calculated from the reported maximum 40 mdeg of CD see figure below). In contrast we report **g-factors up to 1.5**.

Figure S16. Scanning electron microscopy (SEM) images and circular dichroism (CD) spectra of various plasmonic architectures constructed using spherical gold colloids. G-shape values are extracted from ACS Materials Lett. 2021, 3, 3, 282–289, the rest were prepared for this response. Reproduced with permission from Large-scale soft-lithographic patterning of plasmonic nanoparticles. Chiang, N., Scarabelli, L., Vinnacombe-Willson, G. A., Pérez, L. A., Dore, C., Mihi, A., Jonas, S. J., Weiss, P. S. *ACS Mater. Lett.* 3, 282–289 (2021). Copyright 2021 American Chemical Society.

In sum, the templates used until now lacked a clear design that produced a relevant chiroptical signal. Our triskelion array combines the optical modes sustained by the chiral object and those of the lattice to produce g-factors up to 1.5 as demonstrated in our work.

Third and final point. The reviewer mentions that most of these results were expected. However, as we show here, our geometries were engineered to show strong CD values, outperforming other chiral geometries (See new supplementary Fig. S16). Also, we showed that colloidal based triskelion’s outperform solid metal bulk counterparts, which was surprising. We show EELS measurements of a chiral plasmonic array, which to the best of our knowledge is one of the first measurements of this kind of structure. **Finally**, we show how to harness the chiroptical response of the arrays by placing a non-chiral dye on top and producing photoluminescence dissymmetry values ($g\text{-lum}$) of up to 0.55, which exceeds orders of magnitude what is typically observed for chiral dyes (see Arrico et al. *Chem. Eur. J.* 2021, 27, 2970–2934) without altering the brightness of the material.

In sum, we are convinced that our work represents a significant advancement over the state of the art in chiral plasmonics. Our work will enable many groups to endow chirality to a wide variety of colloids (shown here for Au and Ag, but applicable to other materials) and to harness these chiroptical responses to produce highly efficient sensors or chiral light emitters using a simple and scalable approach. In the following table we collect some of the works reporting

significant *g-factor values* in the field of chiral plasmonics to provide additional context for our work.

Table: summary of some of the most representative literature in the field of chiral structures based on metal colloids.

	Material / nanoparticles	Methods	CD (mDeg)	g factor	Reference
1	Gold helicoid nanorods	micelle-directed growth method		0.2	Science 368, 1472–1477 (2020)
2	Gold nanorods (NRs) with human islet amyloid peptides (hIAPPs)	the coassembly of NRs and hIAPPs	2000	0.13	Science 371, 1368–1374 (2021)
3	Gold helicoid nanocubes	Peptide-sequence-specific interactions		0.2	Nature 360, 556 (2018)
4	Chiral gold nanoplates	seed-mediated colloidal synthesis		0.02	Adv. Optical Mater. 2023, 11, 2300037
5	Bichiral plasmonic Nanoparticles	seed-mediated chiral growth method	520	0.011	ACS Nano 2022, 16, 19174–19186
6	Fourfold Twisted Nanoparticles	seed-mediated chiral growth method		0.106	Adv. Mater. 2023, 35, 2208299
7	Chiral Au NRs with five-fold rotational symmetry	a modified seed-mediated growth method		0.26	Angew. Chem. 2023, 135, 202312615
8	Gold spiral structures	nanoparticle chemical lift-off lithography	40		ACS Materials Lett. 2021, 3, 282–289
9	DNA origami 24-helix bundles and gold nanoparticles	Hybridization method of modified gold particles and DNA structure	190		Nature 483, 311–314 (2012)
10	Single-helical gold nanoparticle superstructures	Peptide-Directed Assembly		0.04	J. Am. Chem. Soc. 2016, 138, 13655–13663
11	Chiral plasmonic nanochains	Chiral assemblies of gold nanorods induced by self-associating peptides	100		ACS Nano 2017, 11, 3463–3475
12	Gold nanorods, poly(methacrylate hydroxyethyl-3-indole propionate) and poly(2-hydroxyethyl methacrylate)	Syndioselective reversible addition-fragmentation chain transfer polymerization	300		ACS Nano 2019, 13, 1479–1489

In this work we report CD **7500 mDeg** and **g-factor up to 1.5** and **glum of 0.55**.

- In the SI under the “synthesis of gold nanorods” section, CTAB is described as a solvent which is incorrect as it is a solid at the temperatures described. I’m fairly certain it is being used in an aqueous solution, which should be specified.

Authors: We thank the reviewer for pointing out this mistake. The term "solvent" has been corrected to "surfactant" in the Supporting Information.

- Table S2 describes hypochloric acid as HClO but should be HClO. In addition the methods describe using sodium hypochlorite which is different

Authors: We appreciate the reviewer's careful review. We apologize for the error in Table S2. The compound used in the experiments is sodium hypochlorite (NaClO). Table S2 in the supporting information has been updated to reflect this change.

Reviewer #3 (Remarks to the Author):

In this manuscript Mihi and co-workers create chiral plasmonic lattices from the ‘most common metal colloids’. Plasmonic structures offer a new approach to boost chiral light-matter interactions and achieve strong dissymmetry factors, but sometimes suffer from complicated fabrication requirements and have a limited spectral range. Metal colloids offer the potential to operate in the visible regime, with scalable fabrication processes and often strong chiroptical phenomena. Hexagonal arrays of metal colloids can achieve strong g-factors under normal incidence, and even stronger from different viewing angles. They go on to show that placing an achiral NIR dye on top of the plasmonic array can trigger CPL emission from the dye, where the collection angle impacts the sign/magnitude of the chiroptical response. The article is interesting, well-written and thorough. I am not sure it is appropriate for the broad readership of Nature Communications – similar work has ended up in Scientific Reports – but the science is sound. A few small points below.

Authors: We thank the reviewer for recognizing the interest of our work.

Figure 2 in the SI is difficult to read and could do with a legend (e.g. saying CD: black line, LD: red line, transmittance: dashed blue line). What happens to the optical response if you stack two layers of triskelions?

Authors: We appreciate the reviewer's valuable suggestion. We have modified Figure S2 by incorporating the suggested legends for a better understanding of the data.

Following the reviewer's query, we fabricated bilayer structures incorporating an epoxy SU8 interlayer between two stacked triskelion layers. As depicted in Panel A of Figure R2, the fabrication process involved: (i) assembling nanoparticle arrays of L-triskelions on a glass substrate, (ii) coating the array with an SU8 layer using spin coating, and (iii) assembling a second layer of either L-triskelions or R-triskelions atop the SU8. Panels B and C present the transmittance spectra for each step. The initial array exhibited a prominent resonance at 750 nm. The coating with the SU8 layer significantly modified the optical response and a sharpening of the resonance due to its refractive index matching properties. Finally, adding the second nanoparticle layer resulted in a decrease in transmittance (since there is more metal). Overall, the chiral response is affected as shown in Panels D and E of Figure X, however we did not observe a relevant increase in the g-factor in this case. Here we show the capability to create complex structures with our technique that can be explored in future works.

Figure R2. Schematics of the fabrication process and optical characterization of stacked plasmonic triskelions arrays. A) The fabrication step shows: i) the first triskelion array, ii) triskelion arrays covered with SU8 layer. iii) the second triskelion arrays assembly. B, D) Optical measurements from the sample (L-triskelions/SU8/L-triskelions) C, E) and the sample (L-triskelions/SU8/R-triskelions). The triskelion arrays were made from colloids of Ag 27nm.

Why do the left- and right-handed systems not have equal and opposite optical responses (Figure 3B, D and F)?

Authors: Whereas it is expected that enantiomorphs in **experimental** plasmonic chiral systems may exhibit responses that are not perfectly antipodal, due to unavoidable imperfections between different samples, the mirror-image symmetry observed in the chiral response of a left- and right-handed enantiomer pair (extracted from main text **Figure 3**) is striking. This symmetry encompasses the spectral distribution of modes, overall shape, and relative intensities.

Figure R3. g-factor spectral response of 27nm Au metal colloid (L-, R-) triskelion patterns (from Figure 3)

Why do the Ag nanospheres generate stronger chiroptical responses than the Au nanospheres? Can the authors comment on why the line shape is so different for the two metals (Ag, 3B, seems to be dominated by absorption of one handedness, whilst Au, 3D, looks more balanced)?

Authors: As reported in other plasmonic systems, such as SERS, silver-based structures exhibit a more pronounced spectroscopic response. This characteristic is related to the lower dielectric damping of silver, which translates into sharper localized surface plasmon resonances and more intense local electric fields in the nanoparticles. It is likely that similar mechanisms are governing the chiral response of the structures presented here, originating the observed differences.

The authors should include a photograph/ diagram showing how the angle resolved characterisation (Fig 4, SI Section S9) were performed, as it would be clearer what a 'negative AOI' etc is.

Authors: We appreciate the reviewer's comments regarding this matter. We have revised Figure S17 in Section 8 by including indications in the panel to provide a clearer explanation of the AOI.

The authors should label the y-axis on Figure 5A g-PL, as they have previously used g-factor for g-abs. The angular resolved g-factor spectra are quite confusing, particularly how there is strong g-PL even when there is negligible difference in the red and blue curves (e.g. Fig 5C-E, Fig5F-H). Have the authors measured these spectra more than once, are these results/calculations reproducible?

Authors: We thank the reviewer for these comments. We would like to clarify that the spectra shown in Figure 5A corresponds to g-factor (not to g-lum) obtained from triskelion arrays coated with a SU8 layer. We wanted to show the influence of this coating on the g-factor of the arrays before moving to the PL. Note that the g-factor maximum is located in the 800-900 nm spectral region.

Regarding the obtained g-PL values (in our work we use the notation g_{lum}), in Figure 5 panels D and G the difference between the blue and red curves is very small, and consequently, as shown

in the gray line of these panels, the g-lum is very small. However, for panels C, E, F, and H, a significant difference can be observed between the LCP (red) and RCP (blue) luminescence, resulting in g-lum values between 0.2-0.5. Please note that most dyes show $g_{lum} \sim 10^{-3}$ (doi: 10.1002/chem.202002791). We also demonstrated this effect for Rhodamine B in the supporting information Fig S32. We have rewritten this part to improve how this figure is described.

The g-factor spectra (e.g. Figure 3D and 3F) are very noisy despite the strong dissymmetric response. Can the authors comment on why?

Authors: Increased noise is evident in Figures 3D and 3F for wavelengths exceeding 1100 nm. This is attributed to the use of a different detector in this spectral region compared to the 400-1100 nm range, as detailed in Section 7 of the Supplementary Information.

The SI contains a lot of data which are barely discussed. For example, the main manuscript mentions a “sign-flip of the g-factor”, but it is not clear in the text.

Authors: We thank the reviewer for pointing out this. The text mentioning “sign-flip of the g-factor” has been revised

Given the strongly angle-dependent response, how do the authors propose this can be used in a technological context?

Authors: We thank the reviewer for their interest in potential applications in a technological context.

As a proof of concept demonstrating the seamless applicability of our methodology in optoelectronic devices, chiral arrays were integrated into a photodetector to achieve differential current responses to circularly polarized light. A commercial silicon photodiode (Garosa 2DU2, 3x3 mm² active area) was modified by fabricating chiral arrays on its surface following standard protocols outlined in the manuscript. Silver nanoparticles (27 nm) were assembled to form an array of L-triskelions. The resulting iridescence on the photodetector's active surface as are shown in the photos (Figures R4A, B) confirmed successful assembly. Furthermore, electron microscopy (Figure R4C) revealed a well-defined hexagonal array of triskelions.

The modified photodetector was electrically characterized using a supercontinuum laser (Fianium) as the light source, a linear polarizer, and a quarter-wave plate to generate circularly polarized light of both handedness. Photocurrent was measured in the 500-800 nm wavelength range. The differential photocurrent of the chiral-array-modified photodetector is shown in Figure R4D. These results demonstrate that the integration of plasmonic structures using template assisted self-assembly enables the fabrication of photodetectors with selective responses to incident circular polarization handedness. The methodologies reported in this manuscript are promising for the integration of soft lithography process in the development and miniaturization of technological devices.

This is just an example of the ease in which these structures can be implemented into many devices opening the door for fully exploiting chiral nanophotonics in commercial applications.

Figure R4. Characterization of chiral array-integrated photodetectors. (A, B) Macroscopic images of silicon photodiodes ($3 \times 3 \text{ mm}^2$) modified with template-assembled L-triskelion arrays composed of 27 nm silver nanoparticles. (C) Low-magnification scanning electron micrograph of the triskelion array on the photodiode surface. (D) Differential photocurrent ($i_{\text{LCP}} - i_{\text{RCP}}$) of the modified photodiode as a function of wavelength (500–800 nm), demonstrating selective response to circularly polarized light.

The references seem to omit the work of nearby researchers, e.g. <https://www.nature.com/articles/s41598-021-03908-2>.

Authors: We thank the reviewer for pointing this out. We have included this interesting theoretical work in our list of references.

In conclusion, this is an interesting paper which uses a simple, scalable strategy to achieve strong g-factors. I am still not convinced it will appeal to the multi-disciplinary readership of Nature Communications, nor how translatable the approach is to technological contexts/device architectures.

Authors: As mentioned above to the previous reviewer comments, we are convinced that our work represents a significant advancement over the state of the art in chiral plasmonics. Our work will enable many groups to endow chirality to a wide variety of colloids (shown here for Au and Ag) and to harness these chiroptical responses to produce highly efficient sensors or chiral light emitters using a simple and scalable approach, potentially impacting different research fields. Furthermore, we have shown how this can be implemented in photodetectors. Furthermore, to give an example of the remarkable technological relevance of our structures, we point out to the

recent review of chiral LED technology published in Nature Photonics this year (*Nat. Photon.* 18, p 658 (2024)). The authors conclude that the development of a chiral LED technology depends on finding materials that sustain high circular dichroism values (**glum >0.5**) while maintaining intense photoluminescence (PL) and that can be easily tuned across the visible spectrum. Finding chiral emitters fulfilling all three requirements is very challenging (**glum in most chiral fluorophores is typically $\sim 10^{-3}$**), and chemically modifying the material often severely diminishes the photoluminescence. In our manuscript we show **glum up to 0.55 for both dyes Rhodamine B and the NIR 140 clearly pointing at the potential of this approach to endow chirality to otherwise non chiral emitters.**

In sum, our manuscript contains EELS data of chiral geometries, g-factors up to 1.5 for Au and Ag colloids of different sizes, g-lum values of up to 0.55 for two different dyes, numerical simulations rationalizing the interplay of the optical modes sustained by the chiral object and those of the lattice leading to these results, and a full angular optical characterization of the samples. We strongly believe that the magnitude and the extensive characterization of our structures provides enough merit to be considered for publication in Nature Communications.

Reviewer #4 (Remarks to the Author):

Qi et al. introduce a process to fabricate complex, chiral plasmonic lattices from simple building blocks. The authors use nanoimprint lithography to create chiral templates in a PDMS film, into which they accurately deposit simple, spherical (and rod-shaped) nanoparticles. As a result, complex, chiral lattices arise from simple building blocks. The authors explore their chiroplasmonic properties, show that the lattice resonances can be tailored via the particle size and demonstrate high chiral signals at the spectral positions of these resonances. Finally, they demonstrate that the chiral signature of the created metasurfaces can be used to manipulate light emission from fluorescent dyes in proximity. The manuscript demonstrates a simple approach to highly complex, functional metasurfaces with an attractive combination of a scalable process, tailored chiroptical properties and an interesting application in light emission manipulation. I believe that the manuscript can be of interest to the broad readership of nature communications with some additional clarifications and revisions, as detailed below.

Authors: We are very grateful to the reviewer for the kind words towards our work.

Introduction:

the overview of literature reports could be a bit more detailed to outline the field. The authors mention some examples of top-down (nanofabrication) strategies (e.g. Refs.11-12), and chiral, dispersed colloids (e.g. Refs 27-28) to make their case that simpler strategies are useful (to which I agree). To provide a more complete picture of the field, the authors should consider adding some seminal reports for top-down lithographic approaches – which essentially paved the way for further developments of simpler strategies, such as Liu et al., Nature Photonics 2009, 3, 157 (which paved the way for the field), or Kim et al., Nature 2022, 612, 470 (which developed sensors based on such resonances). In addition, there are several directions in the direction of simpler approaches using self-assembly that may be worth discussing to set the context for their manuscript, including colloidal lithography (Goerlitzer et al., Adv. Mater 2020, 32, 2001330), induced chirality from achiral plasmonic materials (e.g Maoz et al., Nano Lett. 2013, 13, 1203, Goerlitzer et al., ACS Photonics 2023, 10, 1821), or the stacking of achiral objects into 3D chiral structures (e.g. Probst et al, Nature Materials 2021, 20, 1024).

The authors may also want to give some credit to previous work on nanoimprint lithography, which they use in their manuscript without much detail on the fabrication process. Briefly introducing the technique with references would help a reader get access to the framework and set the work in context of previous approaches; including work of the authors themselves (e.g., Kraus et al., Nature Nanotech. 2007, 2, 570; Scarabelli et al., Acc. Mater. Res. 2021, 2, 9, 816; Hanske et al., Nano Lett. 2014, 14, 12, 6863, Matricardi et al. ACS Nano 2018, 12, 8, 8531)

Authors: We thank the reviewer for pointing this out. We have expanded our introduction to make a more comprehensive representation of the state of the art including all these works.

Structure Fabrication:

Why is the g-factor enhanced in the simulations of nanoparticle-based triskelion as compared to bulk gold ones (Figure 1C, D)?

What is the impact of structural imperfections, such as missing arms of the triskelion, particles deposited aside, missing particles, or polydispersity? From the SEM images, the fabrication seems to be of very high quality, but such effects will be found in such fabrications in some areas – especially when considering scaled fabrication.

Authors: We are grateful to the reviewer for raising these important points. In response to the first question, our electrodynamic simulations of triskelions constructed from gold nanoparticles and bulk gold demonstrate that both configurations exhibit comparable chiral response intensities (higher for nanoparticles triskelions), although their spectral profiles differ significantly (Fig R1). These distinct behaviors can be ascribed to the intrinsic properties of nanoparticles, including pronounced plasmonic resonances in the visible spectrum and enhanced extinction efficiency, as well as the strong near-field coupling between individual nanoparticles, facilitating the excitation of chiral modes. Regarding the effects of imperfections such as missing arms, random particles, and size variations, these factors generally lead to a reduction in the measured g-factor. For example, particles located outside the triskelion structure contribute to the overall extinction but do not participate in the chiral response. As a result, the denominator in the g-factor equation increases while the numerator remains unchanged, causing a decrease in the calculated g-factor. To mitigate these effects, substantial efforts have been invested in refining our experimental protocols to produce higher-quality arrays as shown in the additional SEM images in Figs S6-S14.

Optical characterisation

In Figure 4, the authors show the spectral properties of triskelia arrays made of different particle building blocks. They assign the peaks between 600-800nm to the collective lattice resonance, and the higher energy features (400-600nm) to the LSPRs. If the collective resonance is caused by the triskelia themselves (which are formed by many particles in close proximity), why do they shift with the primary particle size? Would one not expect the local mode to change, while the array properties remain similar as the individual triskelia act as a single entity – or is this picture too simplified?

The fluorescence of the pure Rhodamine B dye is in the range of 600nm-700nm (as the authors state in their manuscript). Why is the Circularly polarized photoluminescence measured when the dye was added as a superstrate to the triskelia arrays measured between 800nm-900nm (Figure 5)?

In Figure 5, the authors show the chiral properties (g-factor) of three different Ag NP samples, but in C-E and F-E they compare silver and gold particles with different sizes? Would it not be clearer to show the spectra of the arrays used for the PL studies in Figure 5a to allow a clear correlation between resonance position and PL enhancement?

Authors: We appreciate the reviewer insight regarding the optical characterization.

As the reviewer correctly noted, this system exhibits lattice modes arising from the interactions of triskelions within the hexagonal lattice. Additionally, localized surface plasmon resonances (LSPRs) associated with each triskelion are present at lower wavelengths. The LSPR mode of the triskelion-shaped nanoparticle cluster is NPs size-dependent due to near-field coupling between the nanoparticles. This coupling is influenced by both particle size and interparticle distance. Consequently, since the lattice mode is a result of the hybridization of local modes (LSPRs) of the triskelions and the diffraction modes of the lattice, its characteristics are dependent on the size of the constituent nanoparticles. In response to the reviewer's second comments, we have to clarify that the dye used for the experiments shown in Figure 5 is indeed IR140. This choice is justified by its emission in the 800-900 nm range. Main text: “we mixed the resist with the IR140 fluorophore and used a supercontinuum laser centered at 760 nm as excitation”. In order to mitigate potential misunderstandings, the text describing the experimental procedure has been revised.

Regarding the reviewer's third comment, we chose to show in Figure 5A the spectra of the arrays at the angles where PL was measured to allow comparison of between them, leaving us room to

present the raw PL data measured and the corresponding glum in each case. This is complemented by Figure S33, which shows the chiral characteristics of these samples, and by Figure S27A where the energy at which IR140 emits light is highlighted using a dash line.

REVIEWER COMMENTS

Reviewer #1 (Remarks to the Author):

Recommendation: Accept without further revision

The author responded all our queries and all the reply from the author is satisfactory. Therefore, I recommend the manuscript to publish in its current form.

Authors: We are very grateful to the reviewer for the time invested revising our manuscript and the kind words towards our manuscript.

Reviewer #3 (Remarks to the Author):

The authors have provided adequate responses to the Reviewer's queries.

While they have clarified some sections, they have not taken on board all the changes. For example, in Figure 5, they use the labels "g-factor" and "g-lum". Both are dissymmetry (g-factors). They should indicate that one is for absorption and the other for luminescence on the legend, and the excitation wavelength for luminescence should be included in the caption.

Authors: We use g_{LUM} and **g-factor** as figures of merit to quantify the dissymmetries found in the photoluminescence and/or the absorption following many works in the literature. For instance:

1. Jeanne Crassous, Matthew J. Fuchter, Danna E. Freedman, Nicholas A. Kotov, Jooho Moon, et al.. Materials for chiral light control. **Nature Reviews Materials** volume 8, pages365–371 (2023)
2. Furlan et al. **Nature Photonics** volume 18, pages658–668 (2024)

Both **g-factor** and **g-lum** are described in the manuscript and SI. Specifically:

- Lines 36-39 (Main): "Circular Dichroism (CD) quantifies the difference in absorption between left- and right-handed light when passing through chiral media. These CD values can be expressed normalized to the total absorption, resulting in a well-known parameter: the g-factor or Kuhn's dissymmetry factor.^{10"}
- Lines 292-293 (Main): "figures of merit photoluminescence dissymmetry factor (g_{lum})"
- Supporting Information **Section 8** and equations 1-3.

However, we agree with the reviewer that some readers might find both terms confusing and we have included explicitly the equations for the g-factor (page 2) and g_{lum} (page 13) in the main text. We have also followed their suggestion and improved the clarity of the caption of **Figure 5**, which now reads:

Fig. 5. Circularly Polarized Photoluminescence (CPL) measured in Ag and Au triskelion arrays coated with IR140 doped resist. (A) Maximum g-factor (maximum differences in the extinction for left and right-circularly polarized light) measured from colloidal Ag arrays made with colloids of 27, 32 and 36 nm, covered with a thin layer of SU8 resist ($n \approx 1.6$) and collected at 21°, 31° and 42° angle of incidence. (B) Scheme of the resist coated nanoparticle arrays illustrating the different angles of collection at which the PL was collected. (C-H) Raw photoluminescence (PL) spectra measured under 760 nm excitation, represented by red and blue curves for left- and right-circularly polarized emission, respectively, and the corresponding photoluminescence dissymmetry factor g_{lum} quantifying differences in emission between the two polarizations (grey lines). PL and g_{lum} were collected from 36 nm Ag triskelion arrays at angles of -20° (C), 0° (D), and +20° (E), and from 54 nm Au triskelion arrays at angles of -10° (F), 0° (G), and +10° (H).

We thank the reviewer for suggesting the addition of the excitation wavelength to the figure caption.

The new diagram in Figure S17 is useful to explain the AOI, and much appreciated by the reviewer.

Authors: We thank the reviewer for improving the clarity of the work.

The authors have included some 'chiral photodetectors', where they coat conventional silicon photodetectors with the metasurfaces, in an attempt to show technological applications. These

data do not make it into the manuscript/SI, and I am a little confused. What is the coverage of these devices? From the accompanying photograph, it looks relatively poor (<40%?). It would be good to put the accompanying g-abs so we can track whether the peaks in g-ph correspond to peaks in absorption dissymmetry. Where is the laser spot focussed on the photodetectors?

Authors: In the previous revision, the **reviewer 3** requested proof that our chiral arrays could be implemented in optoelectronics. In our opinion, our manuscript served as introduction of these 2D chiral arrays made of achiral colloids and showing integration in devices would fall outside of the scope of the present work. Still, we decided to provide evidence to the reviewer that our fabrication procedure facilitated seamless integration in devices. To do so, we intentionally covered half of a silicon photodetector with our 2D chiral array and characterized the patterned region focusing our laser in the patterned area versus the non-patterned area, to illustrate how the patterned region was now sensitive to polarization. This characterization and the experimental details can be found in the additional document entitled “*For review only - photodetectors study*”.

This experiment was made in good faith to illustrate the potential of the fabrication methodology and the great performance of the chiral geometry. However, we did not include this new experimental data in our manuscript as it falls beyond of the scope of the current work. While the integration of our chiral in optoelectronics is an exciting avenue of prospective research, there is much design, optimization and characterization to be performed. While we acknowledge the value of such an investigation, it represents a substantial independent research effort rather than a refinement of the current study. We believe that our manuscript, as it stands, provides a comprehensive and robust exploration of the key findings within the intended scope of the study.

In summary, while the authors have boosted the g-factors of chiral metasurfaces, the paper still does not contain novel scientific developments/technological demonstrations, and is more appropriate for another, more specialized, journal.

Authors: We respectfully disagree with the reviewer on this point. In response to **Reviewer 3** queries (July 20th, 2024), we have demonstrated the fabrication of multilayered chiral structures (*not demonstrated with any other methods so far, and available in the supporting information section 7*) and the successful implementation of our chiral arrays in photodetectors (*for review only*). In our manuscript, we have achieved record-breaking chirality values with these structures, which have been acknowledged by the other reviewers and even **Reviewer 3**. Importantly, we demonstrated that these high chirality values can be realized using a scalable approach and we go as far as to demonstrate enhanced circularly polarized photoluminescence of a conventional dye beyond the levels observed in chiral dyes (*typically chiral dyes show g_{lum} of 10^{-3} while we show 0.4*). Therefore, we believe our findings represent a significant advance in chiral plasmonics and merit publication in *Nature Communications*.

Reviewer #4 (Remarks to the Author):

The authors have adequately clarified my remarks on the initial submission. I believe that the combination of a comparably simple fabrication process (soft lithography using simple building blocks vs. synthesis of complex-shaped particles) with a surprisingly strong performance (i.e. better than bulk metal films) and a thorough characterization of structure and optical properties makes this manuscript valuable to a broad scientific audience – as targeted by nature communications.

Authors: We are very grateful to the reviewer for the kind words towards our work.

As a minor remark, I believe that the answer the authors provided to my remark on the influence of structural imperfections (which will increase extinction without contributing to chiral signals) may be mentioned in the main text – simply to describe a potential disadvantage of the system along all the advantages the authors already introduce.

Authors: We are thankful to the reviewer for noting this out. We have happily included said discussion in the main manuscript page 6, which now reads:

The SEM images in Fig. 2 illustrate the quality of the 2D arrays of triskelia achieved with the different colloidal inks. The colloid concentration in each case was optimized to produce clean areas of well-defined triskelia (Fig. S4). **It is also worth mentioning the effects of imperfections such as missing arms, random particles, and size variations. These factors generally lead to a reduction in the measured g-factor. For example, particles located outside the triskelion structure contribute to the overall extinction but do not participate in the chiral response. As a result, the denominator in the g-factor equation increases while the numerator remains unchanged, causing a decrease in the calculated g-factor. To mitigate these effects, substantial efforts have been invested in refining our experimental protocols to produce high-quality arrays as described in the supporting information section 6.**